# Identification of *MLH2/hPMS1* dominant mutations that prevent DNA mismatch repair function

Gloria X. Reyes [1], Boyu Zhao [1,2], Tobias T. Schmidt [1,2], Kerstin Gries[1], Matthias Kloor[3,4,5] & Hans Hombauer [1,6✉]

Inactivating mutations affecting key mismatch repair (MMR) components lead to microsatellite instability (MSI) and cancer. However, a number of patients with MSI-tumors do not present alterations in classical MMR genes. Here we discovered that specific missense mutations in the MutL homolog *MLH2*, which is dispensable for MMR, confer a dominant mutator phenotype in *S. cerevisiae*. *MLH2* mutations elevated frameshift mutation rates, and caused accumulation of long-lasting nuclear MMR foci. Both aspects of this phenotype were suppressed by mutations predicted to prevent the binding of Mlh2 to DNA. Genetic analysis revealed that *mlh2* dominant mutations interfere with both Exonuclease 1 (Exo1)-dependent and Exo1-independent MMR. Lastly, we demonstrate that a homolog mutation in human *hPMS1* results in a dominant mutator phenotype. Our data support a model in which yeast Mlh1-Mlh2 or hMLH1-hPMS1 mutant complexes act as roadblocks on DNA preventing MMR, unraveling a novel mechanism that can account for MSI in human cancer.

[1] DNA Repair Mechanisms and Cancer, German Cancer Research Center (DKFZ), Heidelberg 69120, Germany. [2] Faculty of Bioscience, Heidelberg University, Heidelberg 69120, Germany. [3] Department of Applied Tumor Biology, Institute of Pathology, University of Heidelberg, Heidelberg 69120, Germany. [4] Collaboration Unit Applied Tumor Biology, German Cancer Research Center (DKFZ), Heidelberg 69120, Germany. [5] Molecular Medicine Partnership Unit (MMPU), Heidelberg University Hospital and EMBL, Heidelberg 69120, Germany. [6] Zentrum für Molekulare Biologie der Universität Heidelberg (ZMBH), Im Neuenheimer Feld 282, Heidelberg 69120, Germany. ✉email: h.hombauer@dkfz.de

The DNA mismatch repair (MMR) system corrects mispaired nucleotide bases that arise on DNA[1–4]. DNA mispairs occur during DNA synthesis and escape the proofreading action of replicative DNA polymerases; others are introduced by error-prone DNA polymerases at sites of DNA damage or are caused by enzymatic or chemical modifications of nitrogen bases on DNA. Importantly, inactivating mutations (or epigenetic silencing) of key MMR components result in elevated mutation rates and cancer predisposition[5–7].

In eukaryotes, the recognition of mispaired bases is performed by three MutS homologs (MSH) (Msh2, Msh3, and Msh6) that form two heterodimeric complexes, Msh2–Msh6 and Msh2–Msh3 (also referred to as MutSα and MutSβ, respectively), with partially redundant substrate specificity[8].

Additional members of the MMR family are the MutL homolog (MLH) proteins, which are represented in *S. cerevisiae* by Mlh1, Pms1, Mlh2, and Mlh3. Yeast Mlh1 heterodimerizes with other MLH subunits forming MutLα (Mlh1–Pms1), MutLβ (Mlh1–Mlh2), and MutLγ (Mlh1–Mlh3). In humans, MutLα, MutLβ, and MutLγ heterodimers are represented by hMLH1–hPMS2, hMLH1–hPMS1, and hMLH1–hMLH3, respectively.

In eukaryotes, MutLα is critical for MMR function, whereas MutLβ and MutLγ play a limited role in MMR[1,9,10]. In *S. cerevisiae* Mlh1–Mlh2 is recruited to the mispair site and facilitates the MMR reaction in specific situations[11], while Mlh1–Mlh3 acts mainly during meiosis promoting the resolution of recombination intermediates[12,13]. MLH proteins are homologs to the *E. coli* MutL (EcMutL) MMR protein, and share a related structure consisting of an N-terminal domain (NTD) that possesses ATPase activity, an unstructured linker, followed by a C-terminal domain (CTD) that is necessary for the dimerization. The NTDs of MLH subunits can also dimerize forming a ring-like structure that has been proposed to encircle the DNA[3,4]. Importantly, the CTD of MutLα and MutLγ, but not MutLβ, possess endonuclease domains that allow these complexes to nick DNA[14].

After recognition of mispaired bases, the MutS complex promotes the recruitment of MutLα, which introduces a nick into the newly synthesized strand in the proximity of the mispaired base. In humans, MutLα endonuclease activity is stimulated by the interaction with the Proliferating Cell Nuclear Antigen (PCNA)[15] (called Pol30 in *S. cerevisiae*). Next, the DNA fragment containing the mispaired base is excised either by Exonuclease 1 (Exo1) or in an Exo1-independent manner[16]. After excision, high-fidelity DNA polymerases re-synthesize the excised DNA fragment and the remaining nick is subsequently sealed by DNA ligase I[17].

Mutations inactivating key components of the human MMR system (hMLH1, hMSH2, hMSH6, or hPMS2) are responsible for the most common hereditary cancer predisposition syndrome referred to as Lynch Syndrome or hereditary non-polyposis colorectal cancer (HNPCC)[5–7]. Lynch Syndrome patients are at risk of early onset of cancer due to the accumulation of mutations, especially at repetitive sequences causing microsatellite instability (MSI). However, a significant fraction of colorectal cancer patients with MSI tumors do not present mutations or altered protein expression levels at any of the major MMR components[5], suggesting that additional factors may contribute to either MMR function or DNA replication fidelity at repetitive sequences. Along this line, a previous study demonstrated that inactivation of *SETD2*, the histone methyltransferase that promotes H3K36 trimethylation, causes MSI by preventing the recruitment of MutSα to chromatin[18].

Campbell et al.[11] previously showed that overexpression of either *MLH2* or *MLH3* genes in *S. cerevisiae* completely inactivates MMR function, most likely by outcompeting Pms1 for Mlh1 binding, preventing the assembly of Mlh1–Pms1 complexes that are indispensable during MMR. In addition, this work showed that similar to Mlh1–Pms1[19,20], Mlh1–Mlh2 is recruited in vitro by Msh2–Msh6 to DNA containing a mispaired base. Furthermore, Mlh1–Mlh2 forms short-lived nuclear foci that colocalize with Pms1 foci at sites of repair in vivo[11]. Inactivation of Mlh2 in a wild-type (WT) background causes no significant changes in mutation rates[10,11], however, it facilitates MMR when Mlh1–Pms1 endonuclease function is partially compromised[11]. Based on these previous findings, we investigated whether specific *MLH2* mutations could compromise MMR function, potentially by preventing Mlh1–Pms1 complex assembly, inhibiting MMR steps downstream mispair recognition or by alternative mechanisms. Using a genetic screen in budding yeast, we identified a group of dominant *MLH2* missense mutations, all of them affecting residues at the N-terminus of Mlh2 causing an increase in frameshift mutation rates up to 1000-fold in the *lys2-10A* assay in which a MMR defective strain shows 7000-fold higher rates compared to WT. Further characterization revealed that Mlh1–Mlh2 mutant complexes are recruited to mispair sites where they accumulate, acting as roadblocks on DNA preventing MMR. Finally, we showed that one homolog mutation introduced into *hPMS1* (the human homolog of yeast *MLH2*) causes a dominant mutator phenotype in human cells, suggesting that these type of mutations can lead to increased mutagenesis and cancer predisposition.

## Results

**Identification of *MLH2* dominant mutations resulting in a mutator phenotype.** To search for *MLH2* mutations that could compromise MMR function, we screened a low-copy plasmid library of randomly mutagenized *MLH2*, for a dominant mutator phenotype when expressed in a haploid WT strain. The yeast transformants were screened for a mutator phenotype using two frameshift reversion reporters (*lys2-10A* and *hom3-10*) and the *CAN1* forward inactivation assay[21,22]. The *lys2-10A* and the *hom3-10* reporters are sensitive to single nucleotide deletions that occur in a well-defined mononucleotide run, resulting in lysine (Lys+) and threonine (Thr+) prototrophic colonies, respectively. In contrast, the *CAN1* inactivation assay reports inactivating base substitutions, insertions, deletions, and chromosomal rearrangements resulting in canavanine resistance (CanR). After screening ~71,000 transformants, we identified four Mlh2 mutant alleles resulting in a mutator phenotype: *mlh2-S16P, mlh2-S18P, mlh2-P332L*, and the double mutant *mlh2-S16P-D219G* (Fig. 1a). Interestingly, all mutations located at Mlh2-NTD, which harbors the ATPase domain, and three of them (S16P, S18P, and P332L) affected residues that are part of two disordered loops located at the interface between the two NTDs of the dimer[23,24]. Ser16 and Ser18 are part of the loop 1 (L1) (Fig. 1a), an unstructured region that upon ATP binding adapts an ordered conformation, extending away and interacting with the ATP-binding site of the second subunit of the dimer[24] (Fig. 1b, c), while Pro332 is a conserved amino acid located in loop 3 (L3) at the interface with Mlh1-NTD (Fig. 1b, c). In addition we identified mutated Asp219, which localizes in helix αF that is predicted to be part of a linker (residues 211–233) that connects the ATPase domain (residues 20–210) with the rest of the NTD (residues 234–359)[23].

Expression of these *mlh2* mutant alleles (including *mlh2-V15P* and *mlh2-S17P*, that were generated by site-directed mutagenesis) on a low copy plasmid in a WT strain resulted in elevated mutation rates on the *lys2-10A* reporter, compared to strains expressing WT Mlh2 (Supplementary Table 1). In some cases, a small increase in mutation rates was observed with the less sensitive *hom3-10* frameshift reporter, and no major changes

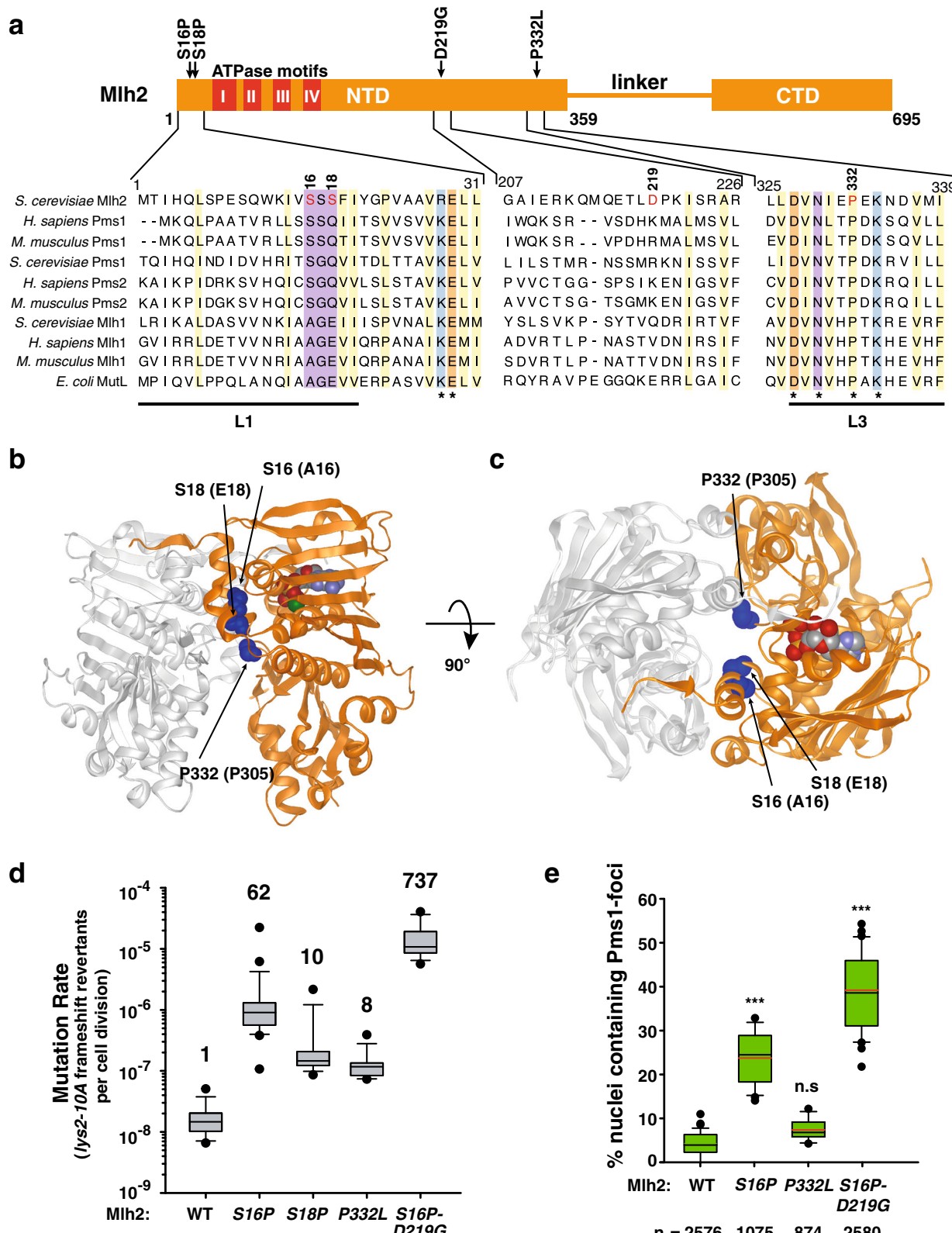

were detected in the *CAN1* forward inactivation assay, likely due to the low sensitivity of this reporter for frameshift mutations (Supplementary Table 1). Similar results were obtained when *mlh2* mutations were introduced at *MLH2*'s chromosomal locus (Fig. 1d and Supplementary Table 2). Furthermore, most of these mutations resulted in an increased abundance of Pms1 foci

(Fig. 1e and Supplementary Fig. 1a), which mark sites of repair and accumulate when downstream steps of the MMR reaction are compromised[20,25,26]. Logarithmic cultures of strains expressing *mlh2* mutations showed similar Mlh2 protein expression levels as a WT strain (Supplementary Fig. 1b), ruling out the possibility that the mutator phenotype was caused by increased Mlh2

**Fig. 1 Identification of *MLH2* dominant mutations causing a mutator phenotype. a** Schematic representation of the Mlh2 protein. Mutated residues are indicated with arrows. Red boxes represent conserved ATPase motifs (I–IV) in MutL homologs. Below, protein sequence alignment of *S. cerevisiae* Mlh2 and MutL homologs in *H. sapiens*, *M. musculus,* and *E. coli* MutL. Residues affected by mutations are marked in red. Conserved hydrophobic residues are shaded in yellow, basic in blue, acidic in orange, and others in purple. * denotes invariable residue across species. Disordered loops L1 and L3 are indicated. **b** Model of Mlh1–Mlh2 heterodimer (Mlh1 in gray, Mlh2 in orange) based on *E. coli* MutL-NTD crystal structure (PDB: 1b62). Mutated residues are indicated as blue spheres. Relevant *S. cerevisiae* amino acid numbers are indicated, followed by the EcMutL homolog residue numbers in parentheses. ADP is indicated in red/gray and $Mg^{2+}$ metal ion in green (only in the Mlh2 monomer). **c** Top view of the heterodimer structure shown in **b**. Models shown in **b** and **c** were made using Protean 3D, Lasergene 15.1, DNASTAR. **d** Mutation rate analysis, represented as box plots with whiskers, using the *lys2-10A* frameshift reversion assay of strains carrying *mlh2* dominant mutations integrated at the chromosomal locus. Numbers on top indicate fold increase in the mutation rate relative to the WT strain. Black dots indicate outliers. **e** Box plots with whiskers indicate the percentage cells containing Pms1-4GFP foci. Black and red lines indicate median and average, respectively. Black dots represent outliers. $n =$ total number of counted cells. ***$p < 0.001$; n.s not significant.

---

protein levels, which would outcompete Pms1 for Mlh1 binding[11].

**Identification of *MLH2* mutations acting as *mlh2-S16P* mutational enhancers**. Among the isolated *mlh2* mutant alleles, the double mutant *mlh2-S16P-D219G* showed the highest mutation rate (737-fold increase in the *lys2-10A* reporter) and the strongest accumulation of Pms1 foci (9 times higher than WT levels) (Fig. 1d, e). Instead, the D219G mutation by itself neither caused a mutator phenotype (Supplementary Table 2) nor resulted in increased Pms1 foci abundance (Supplementary Fig. 1a). These results suggest that S16P and D219G mutations affect different processes of the Mlh1–Mlh2 complex behavior, and the D219G mutation acts as an enhancer of S16P mutator phenotype, causing a synergistic increase in the mutation rate.

To test whether other *MLH2* mutations could further enhance *mlh2-S16P* mutator phenotype, we performed an additional screen where we randomly mutagenized *mlh2-S16P* and searched for transformants that grew in at least two of the mutator reporter plates. Among ~37,000 transformants, we identified six distinct *mlh2-S16P* enhancer mutations: D45N, E99K, F177L, D178N, E216G, T217A and the D219G mutation previously identified. All isolated plasmids contained missense mutations affecting residues at Mlh2-NTD (Fig. 2a, b); most of them (with exception of E99K) were predicted to locate at the surface of the protein. Asp45 is part of the β1 sheet flanking the first ATPase motif (Supplementary Fig. 2a, b), and its side chain is likely exposed to solvent. Glu99 is immersed in a conserved region among MutL homologs, characterized by the "GFRGEAL" sequence, which is part of the ATP-binding motif III[2,23] (Supplementary Fig. 2a). The remaining amino acid substitutions clustered in two regions: Phe177 and Asp178 located along the helix αE in EcMutL structure[23], a region with a relative conservation in hydrophobic residues among MutL homologs (Supplementary Fig. 2a), and residues Glu216, Thr217, and Asp219 clustered in fairly well-exposed linker region with low amino acid conservation (Fig. 2a, b and Supplementary Fig. 2a) that connects the ATPase domain with the rest of the NTD[23].

Yeast strains carrying double *mlh2* mutations showed in average a 12-fold higher mutation rate than strains with the S16P single mutation (Fig. 2c and Supplementary Tables 1 and 2). Analysis of the *CAN1* mutation spectra in these strains revealed that more than 60% of the *CAN1*-inactivating mutations were single nucleotide deletions (and few insertions) mainly at mononucleotide runs (4–6 bases); whereas the WT mutation spectrum was dominated by base substitutions (75%), with a small fraction (16%) of frameshifts (Fig. 2d and Supplementary Table 4). The preponderance of frameshifts over base substitutions in *mlh2* double mutants is indicative of a severe MMR defect, similar as described for an *msh2Δ* strain[8,27]. In line with the effect on mutation rates, strains carrying double *mlh2* mutations (except for S16P-D45N) showed a significantly higher

percentage of cells containing Mlh2 foci compared to the *mlh2-S16P* strain ($p < 0.001$) (Fig. 2e, f).

Strains expressing double *mlh2* mutations showed in average 2–3 times brighter Mlh2 foci than WT (Fig. 2f), which was not caused by an evident increase in Mlh2 (or Pms1) protein levels (Supplementary Fig. 2b). In addition, strains expressing *mlh2-S16P-D219G* allele showed Pms1 foci that in average persisted 8-times longer and were 2.9-times brighter than WT Pms1 foci (Fig. 3a, b, c).

Analysis of Pms1 protein levels throughout the cell cycle revealed a transient S-phase-dependent expression that peaks at 30 min after release from α-factor arrest (Fig. 3d), and is consistent with Pms1 mRNA levels during cell cycle[28,29]. Similarly, Mlh2 protein expression was highest during S phase. Interestingly, strains expressing the *mlh2-S16P-D219G* allele showed for both, Pms1 and Mlh2 proteins, a less-tight S-phase-dependent expression pattern (Fig. 3d). Both proteins accumulated at earlier time points and remained detectable until beginning of G2/M (indicated by the expression of the Clb2 cyclin). These results are in agreement with the more abundant and long-lived Pms1/Mlh2 foci observed in strains carrying *mlh2* mutant alleles.

**The *mlh2-S16P* mutation results in a stronger Mlh1–Mlh2 interaction by yeast two-hybrid**. As Ser16 locates at the NTD–dimer interface (Fig. 1b, c)[23,24], we explored the possibility whether this mutation could have an impact on the interaction between Mlh2 and Mlh1. To test this idea, we took advantage of the yeast two-hybrid (Y2H) system, which has been used to detect the interaction between Pms1 and Mlh1[30,31]. Growth on reporter plates (Trp⁻ Leu⁻ His⁻) (Fig. 4a), revealed a positive interaction between Pms1 and Mlh1, and a very weak interaction between Mlh2 and Mlh1, despite Mlh2's higher expression level (Supplementary Fig. 3a). Remarkably, the Mlh2-S16P mutant protein interacted with Mlh1 much stronger than WT Mlh2 (Fig. 4a), which is likely caused by the Ser to Pro substitution that will affect the orientation of the first N-terminal 16 residues. This result suggests that the S16P mutation is either promoting the association with Mlh1, or preventing the dissociation between Mlh1- and Mlh2-NTDs. Furthermore, strains carrying double *mlh2* mutations caused a similar increase in the Y2H interaction, suggesting that *mlh2-S16P* enhancer mutations are interfering with MMR using a different mechanism.

**mlh2-S16P mutation interferes with Exo1-dependent and Exo1-independent MMR pathways**. To gain further insight into the mechanism how *mlh2* mutations prevent MMR function, we generated double mutant strains carrying the *mlh2-S16P* (or *mlh2-S16P-D219G*) allele and a mutation that either inactivates the Exo1-dependent or Exo1-independent MMR pathways[16]. In agreement with previous reports, inactivation of Exo1 (*exo1Δ*) resulted in a modest mutator phenotype[20,26,32] (Supplementary

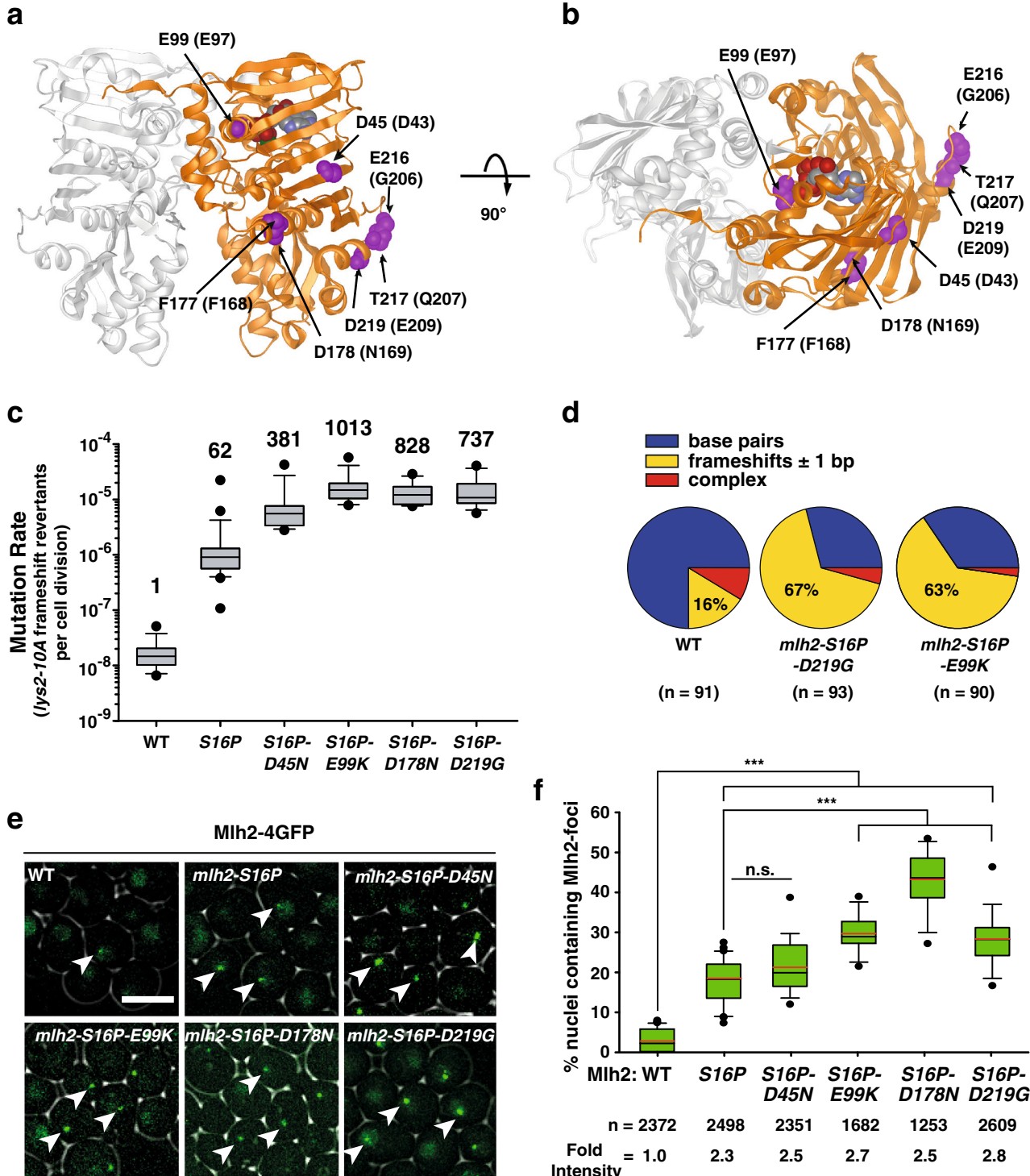

**Fig. 2 Identification of *mlh2-S16P* mutational enhancer mutations. a** Model of the Mlh1–Mlh2 heterodimer structure (Mlh1 in gray, Mlh2 in orange) indicating residues affected by mutations with arrows (purple spheres). Mlh2 amino acid numbers are shown followed by EcMutL homolog residue numbers in parentheses. **b** Top view of the Mlh1–Mlh2 heterodimer shown in **a**. **c** Frameshift mutation rates (*lys2-10A* reporter) of strains carrying double *mlh2* mutations integrated at the chromosomal locus shown in box plots with whiskers. Numbers on top of each box indicate the fold increase in the mutation rate over the WT and black dots represent outliers. **d** *CAN1* mutation spectrum in the indicated strains. Independent canavanine-resistant (Can^R) colonies (*n* ≥ 90 per genotype) were sequenced for *CAN1* mutations. Pie graphs show the relative distribution of identified mutations. **e** Confocal live-cell images of Mlh2-4GFP foci (examples are shown with white arrows) in logarithmically growing cells with the indicated genotype. Bar represents 5 μm. **f** Quantification of Mlh2-4GFP foci in the indicated strains, shown as box plots. The black and red lines indicate median value and average, respectively. Black dots represent outliers. *n* indicates the total number of cells counted per genotype. Fold intensity shows the fold increase of the foci intensity of the *mlh2* mutants over the WT foci intensity. ***$p$ < 0.001.

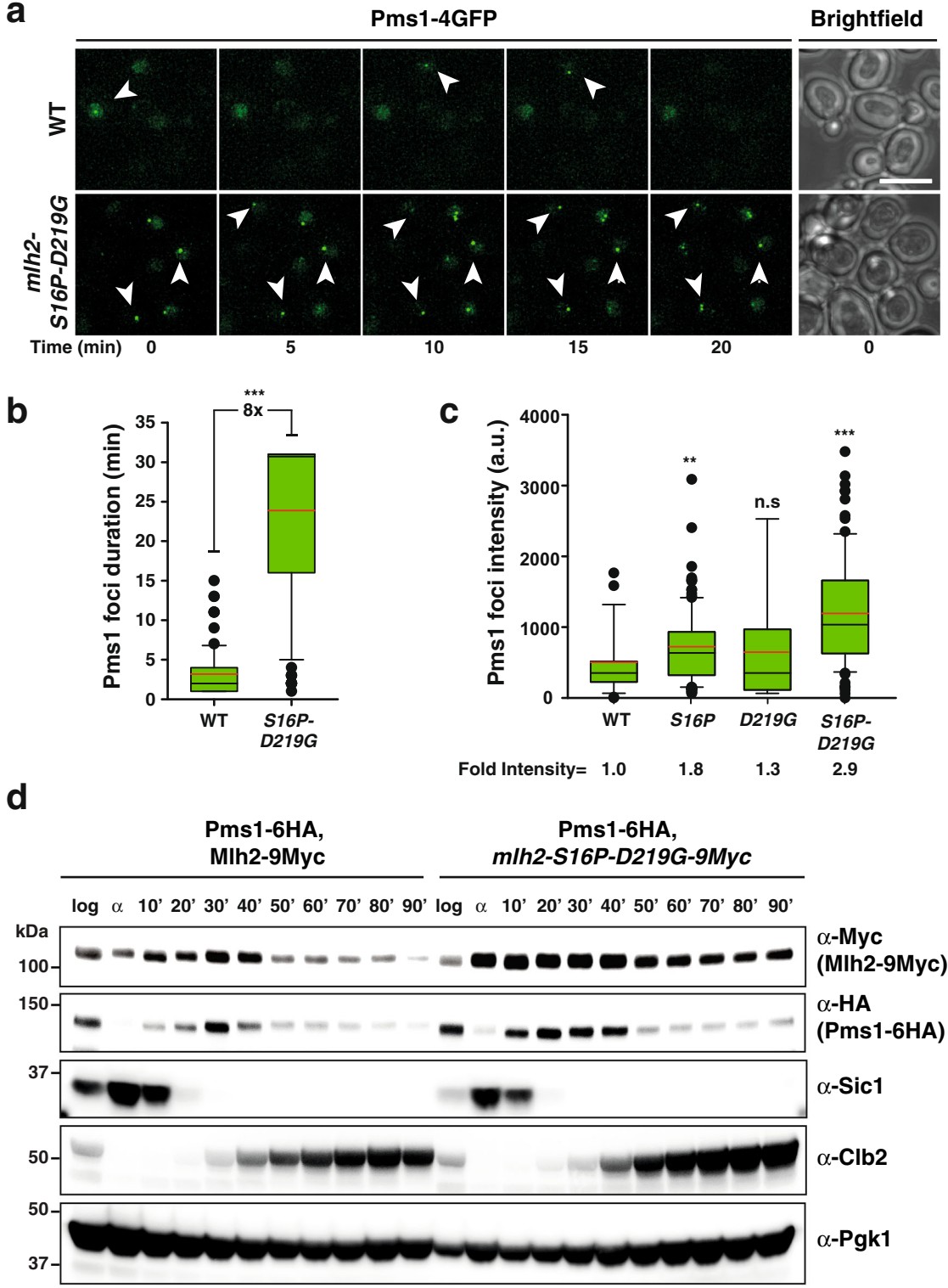

**Fig. 3 *MLH2* mutations result in abundant and persistent Pms1-4GFP foci, and altered Pms1 and Mlh2 expression pattern. a** Representative time-lapse images following Pms1-4GFP foci in the WT and *mlh2-S16P-D219G* double mutant strain, over a 20 min period. Images were captured every minute. Arrowheads follow a focus in time; bar represent 5 μm. Brightfield images were taken at the beginning of the time-lapse. **b** Quantification of Pms1 focus duration in the WT and *mlh2-S16P-D219G* double mutant. **c** Intensity of Pms1 foci was calculated in strains with indicated *mlh2* mutation. Fold intensity shows the fold increase of the foci intensity (averaged value) of the *mlh2* mutants compared with the WT. In **b** and **c**, the data are presented as box plots with whiskers, in which the black and red lines indicate the mean and the average, respectively. Black dots represent outliers. **d** Pms1 and Mlh2 protein expression in the WT and *mlh2-S16P-D219G* double mutant strains analyzed by Western blot. Lysates were prepared with α-factor synchronized cells that were released at different time points, as indicated. Data are representative of two independent experiments, which gave similar results. Sic1 and Clb2 were used as G1- and G2/M-phase markers, respectively. Pgk1 was used as a loading control. ***$p < 0.001$; **$p < 0.01$; n.s indicates not significant compared to WT.

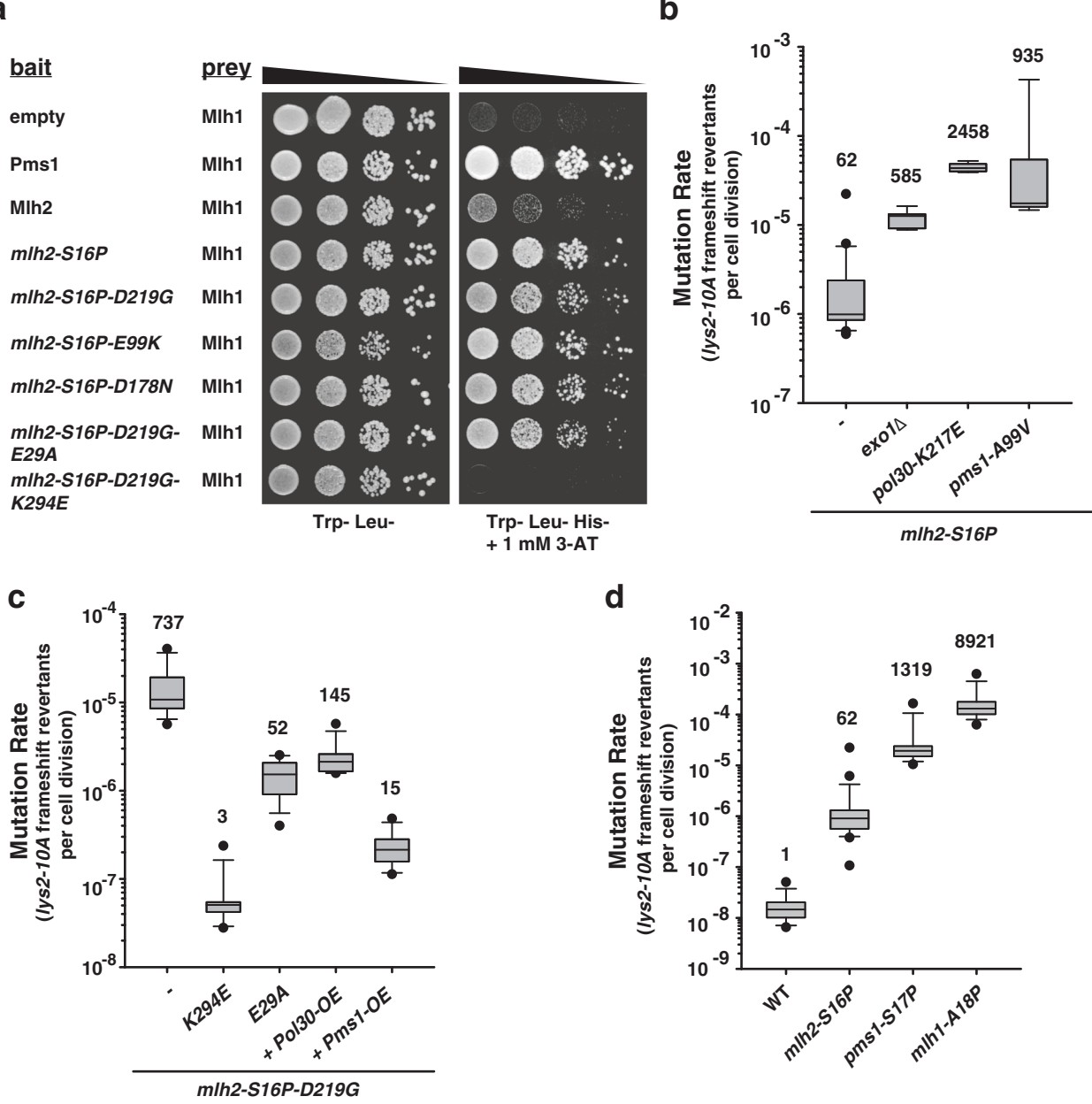

**Fig. 4 *mlh2-S16P* mutation results in an increased interaction with Mlh1 and interferes with Exo1-dependent and Exo1-independent MMR pathways.**
**a** The interaction between Mlh1 (prey) and WT- or mutant Mlh2 (baits) proteins was tested by Y2H. Prey and bait plasmids contain *TRP1* and *LEU2* auxotrophic markers, respectively. Interaction was scored as growth on Trp⁻ Leu⁻ His⁻ +1 mM 3- AT plates. Cells were spotted in serial dilutions on control (Trp⁻ Leu⁻) or reporter plates (Trp⁻ Leu⁻ His⁻ +1 mM 3-AT). Mlh1 and Pms1 proteins serve as positive control for the Y2H interaction. **b–d** Box plots with whiskers showing mutation rates in the indicated strains using the *lys2-10A* frameshift reporter. The numbers on top correspond to the fold increase in mutation rates over the WT. Black dots indicate outliers.

Table 3a). However, *exo1Δ* mutation in combination with either *mlh2-S16P* or *mlh2-S16P-D219G* mutant allele caused a strong increase in frameshift mutation rates (Fig. 4b and Supplementary Table 3a). These results indicate that the *mlh2-S16P* mutation is interfering with Mlh1–Pms1 nicking activity, which normally compensates for the lack of Exo1. Moreover, combining the *mlh2-S16P* allele with mutations that inactivate the Exo1-independent pathway (*pol30-K217E* or *pms1-A99V*) by preventing Mlh1–Pms1 nicking activity also caused a synergistic increase in frameshifts (Fig. 4b and Supplementary Table 3a). Together, these results revealed that the *mlh2-S16P* mutation causes a partial defect in both, Exo1-dependent and Exo1-independent MMR pathways.

The accumulation of Mlh2 foci in strains carrying the *mlh2-S16P* mutant allele, suggests that the *mlh2* mutant complexes are being recruited to mispair sites on DNA. Based on the increased Y2H interaction between Mlh2-S16P and Mlh1, we speculated that the Mlh1–Mlh2 mutant complexes may remain associated with DNA, acting as roadblocks interfering with the MMR reaction. To test if the mutator phenotype caused by *mlh2* mutations requires the interaction of Mlh1–Mlh2 mutant complexes with DNA, we introduced the K294E mutation into the *mlh2-S16P-D219G* allele, that is expected to prevent DNA binding[33,34]. Remarkably, the K294E mutation completely suppressed the mutator phenotype (Fig. 4c and Supplementary Table 3b) and the accumulation of Mlh2 foci, characteristic of the

*mlh2-S16P-D219G* allele (Supplementary Fig. 3b). A similar result was seen after introducing the E29A mutation that prevents MutL-dependent ATP hydrolysis[23] (Fig. 4b, Supplementary Table 3b and Supplementary Fig. 3b). These effects were not caused by a destabilization of the protein, as both triple mutants were expressed at similar levels than *mlh2-S16P-D219G* protein (Supplementary Fig. 3c). In addition, we found out that the K294E mutation, but not the E29A mutation, completely abolished the Y2H interaction between *mlh2-S16P-D219G* and Mlh1 (Fig. 4a and Supplementary Fig. 3a). It is interesting that the K294E mutation also suppressed the dominant mutator phenotype (Supplementary Table 3b) and the Mlh2 foci accumulation (Supplementary Fig. 3b) associated with the *mlh2-S16P-D219G* mutation more efficiently than the E29A mutation. Together, these observations indicate that Mlh1–Mlh2 mutant complexes impart a dominant mutator phenotype, most likely by preventing downstream steps of the MMR reaction.

We also found that overexpression of *POL30* or *PMS1* (by replacement of their endogenous promoters with a strong constitutive *pGPD* promoter), largely suppressed the mutator phenotype and the accumulation of Mlh2 foci associated with the *mlh2-S16P-D219G* allele (Fig. 4c, Supplementary Table 3b and Supplementary Fig. 3b). The suppression caused by Pms1 overexpression (Pms1-OE) is in agreement with the fact that the Mlh2 mutant protein has to compete with Pms1 for Mlh1 binding. On the other hand, the suppression by Pol30 over-expression (Pol30-OE), and the synergistic increase in mutation rates after combining *mlh2-S16P* with *exo1Δ*, both indicates that Mlh1–Mlh2 mutant complexes prevent the interaction between Pol30 and Mlh1–Pms1 that is required for the activation of the Mlh1–Pms1 endonuclease.

It is tempting to speculate that this interaction may also trigger a conformational change in Mlh1–Pms1 (and potentially Mlh1–Mlh2) that facilitates the unloading of these complexes from DNA. This idea is consistent with the accumulation of Pms1 foci in yeast strains carrying Pol30 mutant alleles that are defective in the activation of the Mlh1–Pms1 endonuclease[26], but also with the suppression of *mlh2-S16P-D219G* mutator phenotype and the reduction in Mlh2 foci abundance observed upon Pol30-OE.

As the S16P mutation in Mlh2 disturbs a fairly well-conserved residue among MutL homologs, we asked whether homolog mutations in yeast Pms1 or Mlh1 might compromise their function. Mutation rate analysis in strains carrying the Pms1 homolog mutation (*pms1-S17P*) showed an increased mutator phenotype, whereas strains containing the Mlh1 homolog mutation (*mlh1-A18P*) were completely MMR defective (Fig. 4d and Supplementary Table 3c). The fact that the homolog mutation is more deleterious in Mlh1 than in Pms1, is probably related to the asymmetry of both NTDs, being more critical the Mlh1-NTD ATPase function[31,35]. Moreover, these results suggest that the loop 1 plays a conserved and critical function during MMR among yeast MLH proteins.

In summary, these results support a model in which Mlh1–Mlh2 mutant complexes are loaded on DNA, where they remain associated for a longer period than WT complexes. We propose that these mutant protein complexes act as roadblocks on DNA preventing downstream steps of MMR.

**The *hPMS1*-S14P mutation confers a dominant mutator phenotype in mammalian cells**. Cumulative evidence indicates that inactivation of human PMS1 (hPMS1), the homolog of yeast Mlh2, is not associated with increased mutagenesis or cancer susceptibility[6,7,36]. Still, specific mutations in the *hPMS1* gene could result in mutant protein complexes that may interfere with human MMR, analogous to our findings in *S. cerevisiae*. We used CRISPR-Cas9 gene editing to introduce the S14P point mutation (homolog mutation to yeast *mlh2-S16P*) into *hPMS1* in human HAP1 cells (Supplementary Fig. 4). In addition, we generated an *hMLH1*-knockout (*hMLH1*-KO) HAP1 cell line that was used as MMR-deficient control, as well as a *hPMS1*-knockout cell line (*hPMS1*-KO) that was expected to be MMR-proficient according to previous studies[6,37,38]. The *hPMS1*-S14P mutation did not affect the overall stability of the protein (Fig. 5a). In contrast, inactivation of hMLH1 resulted in a strong reduction in hPMS1 levels, in agreement with the fact that MLH proteins are only stable as heterodimers.

Next, we evaluated the mutator phenotype of HAP1 mock cells (transfected with a plasmid expressing a single guide RNA (sgRNA) targeting the green fluorescent protein (GFP)) and HAP1-mutant cells using the hypoxanthine-guanine phosphor-ibosyltransferase 1 (HPRT1) inactivation assay, which scores for *HPRT1* mutations resulting in 6-TG resistance. Cells expressing a functional *HPRT1* gene, convert 6-TG into toxic nucleotides that are incorporated into DNA, triggering G2/M arrest and cell death. In contrast, cells with a strong mutator phenotype, like MMR-deficient cells, frequently inactivate the *HPRT1* gene and become resistant to 6-TG. Cells carrying the *hPMS1*-S14P mutation were more resistant to 6-TG than HAP1-mock cells, but not as resistant as *hMLH1*-knockout cells (Fig. 5b). In a quantitative colony formation assay at a concentration of 0.8 μM 6-TG, only 15% of mock-transfected cells remained alive, whereas 50% of the *hPMS1*-S14P mutant cells and 82% of *hMLH1*-knockout cells survived (Fig. 5c). On the other hand, HAP1 cells lacking the *hPMS1* gene showed a similar sensitivity to 6-TG as the HAP1 mock-transfected cells (Supplementary Fig. 6), which is consistent with the minor role of hPMS1 in MMR[37,38].

In budding yeast, strains carrying the *mlh1-A18P* mutation were completely MMR defective (Supplementary Table 3c). Similarly, we found that HAP1 cells carrying the *hMLH1-A21P* homolog mutation were as strong mutators as *hMLH1*-KO cells (Fig. 5e, f) and presented unchanged hMLH1 expression levels (Fig. 5d and Supplementary Fig. 5).

In summary, these results demonstrated that specific *hPMS1* mutations (and potentially also homolog mutations in *hMLH1*) can confer a dominant mutator phenotype in human cells.

## Discussion

Here we discovered missense mutations affecting the yeast MutL homolog Mlh2 or its human homolog hPMS1 that result in elevated mutation rates. The isolation of these *mlh2* mutant alleles, together with the previous observation that deletion of the *MLH2* gene in *S. cerevisiae* does not cause a significant mutator phenotype[10,11] indicates that these *mlh2* mutant alleles impart a dominant mutator phenotype by interfering with the function of other more relevant MMR proteins. The identified *mlh2* mutant alleles did not affect Mlh2 or Pms1 protein levels; therefore, the underlying cause of the mutator phenotype is different from the one reported for Mlh2 or Mlh3 overexpressing strains[11]. All the identified mutations affected residues located exclusively at Mlh2's NTD, and can be grouped in two categories: 1) "*mlh2*-mutators", those mutations causing a mutator phenotype by itself (e.g., S16P, S18P, and P332L), and 2) "*mlh2*-S16P mutational enhancers", which are not mutators per se, but caused a synergistic increase in mutation rates in combination with the *mlh2*-S16P allele (e.g., D45N, E99K, D178N, and D219G). Our screen revealed some similarities with a study done in *E. coli* that identified a group of MutL dominant negative mutations, causing a mutator phenotype in the presence of a WT MutL gene[39].

Interestingly, among the identified EcMutL dominant mutations were A16T, A16V, and P305L, which are affecting the homolog residues to Ser16 and Pro332 in Mlh2, respectively. On the other hand, this previous study[39] reported a variety of additional EcMutL dominant mutations that were not found in our screen, despite the fact that our screen was largely saturated as several of the *mlh2* mutations were identified more than once. These differences could be related to the fact that in *E. coli* there is only one

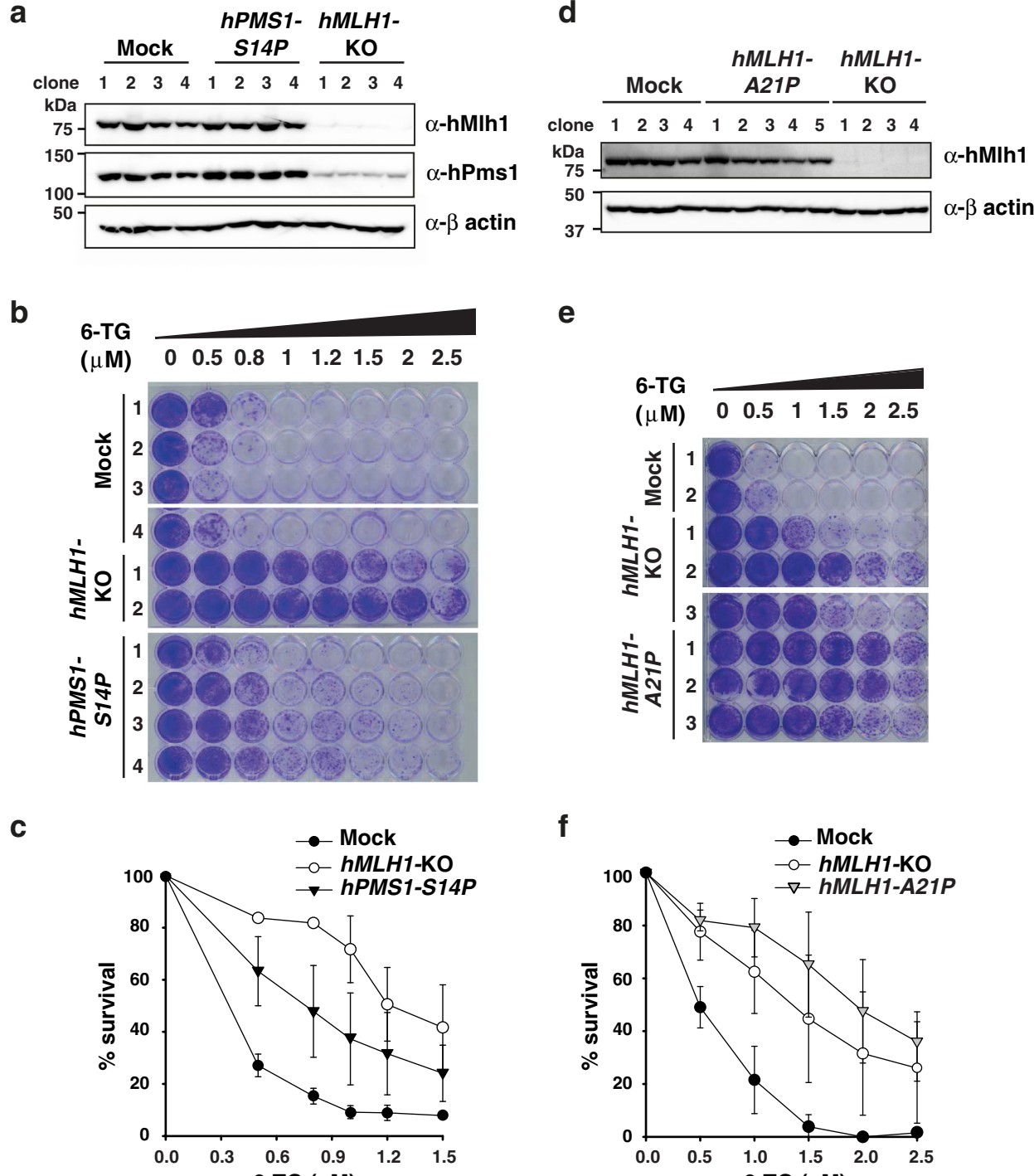

**Fig. 5 *hPMS1-S14P* and *hMLH1-A21P* mutant alleles confer mutator phenotype in human HAP1 cells. a** hPMS1 and hMLH1 expression levels in CRISPR/Cas9-edited *hPMS1-S14P* and *hMLH1*-KO cell lines, determined by Western blot. Actin was used as a loading control. **b** Qualitative mutator analysis showing increased resistance to 6-TG in *hMLH1*-KO and *hPMS1-S14P* cells compared to mock cells. Cells were treated with 6-TG at the indicated concentration for 7 days and were stained with crystal violet. **c** Survival curves for *hMLH1*-KO ($n = 4$, biologically independent clones) and *hPMS1-S14P* cells ($n = 4$, biologically independent clones) based on a colony-formation assay after 12 days of treatment with different concentrations of 6-TG. **d** Western blot analysis showing hMLH1 expression levels in the indicated cell lines. **e** Qualitative mutator analysis (as shown in **b**) for the indicated cell lines. **f** 6-TG survival curves for HAP1-mock ($n = 6$, biologically independent clones), *hMLH1*-KO ($n = 4$, biologically independent clones), and *hMLH1-A21P* ($n = 3$, biologically independent clones), determined as in **c**. For **c** and **f**, error bars represent standard deviation of the mean of the data set.

homodimeric MutL complex, whereas in eukaryotes there are up to three heterodimeric MutL complexes.

In vivo EcMutL exists as homodimer, promoted by the interaction of EcMutL C-terminal domains, while the NTDs can dimerize in response to ATP binding[23]. The association of the NTDs has been proposed to result in a central channel that allows EcMutL to enclose the DNA helix[3,24]. Most likely, this step is a prerequisite for nicking of the newly synthesized strand near the mispair site. Along this line, mutations at the ATPase domain in EcMutL or yeast MutLα cause MMR defects[39,40]. Moreover, a number of Lynch Syndrome patients harbor mutations at, or nearby the ATPase domain (Supplementary Table 5), suggesting that these mutations compromise the ATPase activity preventing MMR function.

Visualization of MutLα complexes by atomic force microscopy revealed that MutLα go through an ATPase cycle, in which ATP binding promotes dimerization of the NTDs and an overall highly compacted state, whereas ATP hydrolysis favors the dissociation of the NTDs and a more relaxed state[41]. The ATP-driven conformational change results in the interaction between the N- and C-terminal domains, potentially influencing the nicking reaction at the MutLα C-terminus. These findings are in agreement with recent single-molecule imaging studies of EcMutS and EcMutL proteins that have shown that both EcMutS and EcMutL, in their ATP-bound state, form sliding clamps that diffuse along DNA to direct MMR[42,43].

According to the EcMutL crystal structure[23,24] (Fig. 1), mlh2 mutations (e.g., S16P, S18P, or P332L) are affecting residues located at two disordered loops at the interface of EcMutL NTDs. These loops that become structured upon ATP binding are predicted to act as conformational switches[23,24,34,44]. Given that the identified mlh2 mutator alleles (S16P, S18P, P332L) have in common a proline substitution, it is expected that these substitutions will drastically change the orientation of loops L1/L3; potentially affecting the ability to sense ATP hydrolysis at the contiguous NTD, which could prevent the dissociation of the NTDs. This prediction is supported by the observation that the mlh2-S16P mutation results in a stronger interaction with Mlh1 by Y2H (Fig. 4a), similar to mutations that prevent ATP hydrolysis in yeast Mlh1/Pms1 subunits[31].

On the other hand, most mlh2-S16P-enhancer mutations (with exception of E99K) are affecting residues located at the surface of the NTD-MutL dimer, suggesting that these residues could participate in transient protein–protein interactions, either with components of the MMR pathway (e.g., Msh2-Msh6 or PCNA) or perhaps, proteins that could be involved in the recycling of MutL subunits. Interestingly, the mlh2-D45N enhancer mutation affects a residue located at the predicted interface according to the EcMutS/EcMutL crystal structure[45], suggesting that the D45N mutation could alter the interaction between Mlh2 and MutSα (or MutSβ). However, this possibility is rather unexpected, since the homolog interaction in humans is predominantly mediated by the hMLH1 subunit[46,47].

Strains expressing mlh2 mutant alleles show increased abundance of Mlh2- and Pms1 foci, similar to mutations preventing Pms1 endonuclease function or Exo1-dependent excision[11,20,25,26]. Furthermore, Pms1/Mlh2 foci were brighter and long lasting, suggesting that the mlh2 mutations result in mutant protein complexes that are loaded on DNA, where they remain associated longer than usual. Accordingly, the mutator phenotype and the increased abundance of Pms1/Mlh2 foci caused by the mlh2-S16P-D219G allele were both suppressed by mutations that either prevent DNA binding (K294E) or inhibit ATP hydrolysis (E29A). Both mutations are expected to prevent the loading of Mlh1–Mlh2 mutant complexes via two different mechanisms; K294E mutation is predicted to decrease DNA binding affinity,

whereas the E29A mutation results in a constitutive ATP-bound closed conformation with dimerized NTDs.

The analysis of Pms1 protein expression throughout the cell cycle in a WT strain, revealed a transient S-phase-dependent expression, which is consistent with Pms1 mRNA levels[28,29]. Similarly, Mlh2 protein expression was highest during S phase. Interestingly, strains expressing mlh2-S16P-D219G mutant allele showed an altered Pms1 and Mlh2 expression pattern throughout the cell cycle. Although it is possible that the Mlh1–Mlh2 mutant complexes may affect PMS1/MLH2 gene expression, it is more likely that these mutant protein complexes are interfering with the degradation of Mlh2 and Pms1. We speculate that Mlh1–Mlh2 mutant complexes are preventing the unloading of Mlh1–Pms1 from DNA, a process that might be somehow coupled to the degradation of Pms1 and Mlh2 subunits, and perhaps Mlh1 recycling. Intriguingly, several components of the ubiquitin-proteasome pathway were previously found associated with hPMS2 and hPMS1 (the homologs of yeast Pms1 and Mlh2, respectively)[48], suggesting that both subunits could be targets of proteasome-mediated degradation, possibly as a mechanism to restrict their availability to the time when the DNA is replicated.

The synergistic increase in the mutation rate observed after combining the mlh2-S16P allele with exo1Δ, but also with mutations that prevent Pms1 endonuclease activity (pol30-K217E and pms1-E99V) (Fig. 4b and Supplementary Table 3a), indicates that the mlh2-S16P mutation partially interferes with both Exo1-dependent and Exo1-independent MMR pathways. On the other hand, the mutator phenotype caused by the mlh2-S16P-D219G allele was suppressed by Pms1 or Pol30 overexpression. Increased Pms1 levels will most likely outcompete mutant Mlh2 for Mlh1 binding, while the suppression by Pol30-OE suggests that Mlh1–Mlh2 mutant complexes interfere with the PCNA-dependent activation of Pms1 endonuclease activity. These genetic interactions are compatible with a model in which Mlh1–Mlh2 mutant complexes act as roadblocks on DNA interfering with PCNA-dependent activation of Pms1 endonuclease and Exo1-dependent excision.

A previous mutational study in S. cerevisiae identified a number of conserved residues located at the surface of Pms1-NTD that when mutated resulted in an elevated mutator phenotype[34]. Some of these mutations decreased DNA binding, whereas others including R212E/K213E did not. Interestingly, Arg212 and Lys213 are located at the helix αF that connects the ATP binding region from the rest of the NTD that corresponds to the same region identified in the present study as one cluster of mlh2-S16P enhancer mutations (D216G, T217A, and D219G). Certainly, this similarity highlights the importance of this linker region during the ATP-binding dependent conformational change, in which several regions of the NTD become ordered and compacted.

Analysis of human cancer genome databases (International Society for Gastrointestinal Hereditary Tumors (InSIGHT), Clinically relevant variants (ClinVAr), and the Catalog of Somatic Mutations in Cancer (COSMIC)) revealed, for all here-identified yeast MLH2 mutations, homolog substitutions in human MLH genes (hMLH1, hPMS2, and hPMS1) (Supplementary Table 5). Among them, hMLH1-A21V (homolog to Ser16 in S. cerevisiae Mlh2) has been found in Lynch Syndrome patients and is classified as pathogenic, while the hMLH1-E23D mutation (homolog to Ser18 in S. cerevisiae Mlh2) as well as the hMLH1-P309L (homolog to Pro332 in S. cerevisiae Mlh2) have been classified as uncertain. The hMLH1-Q48P mutant (homolog residue to Asp45 in S. cerevisiae Mlh2) has been classified as deleterious, as caused protein instability, reduced interaction with hPMS2, and a dominant negative mutator effect when tested in yeast[49]. The mutant variants hMLH1-E102D/E102K (homolog to Glu99 in

Mlh2) had no effect on protein stability but showed ~50% of WT activity in an in vitro MMR assay using mismatched DNA heteroduplex[50].

Interestingly, among the somatic mutations reported in the COSMIC database, we identified one patient with chronic lymphocytic leukemia carrying the *hPMS1-P312S* mutation, which affects a residue homolog to Pro332 (also identified in our Mlh2 screen). However, the pathogenic potential of this *hPMS1* mutation has not yet been investigated. As listed in Supplementary Table 5, additional homolog mutations to the *mlh2* mutator/ *mlh2-S16P* enhancer mutations have been identified in *hPMS2* and *hPMS1* in a variety of cancer types, suggesting that these mutations could result in mutator phenotypes and cancer susceptibility. For some of these human mutant variants there are functional studies indicating an impaired MMR function. However, most of these studies cannot distinguish between loss of function or dominant mutations. Our study points out to specific *MLH2* mutations that confer a dominant mutator phenotype, and predicts that homolog mutations in other MutL homologs may have similar consequences. In part, this idea is supported by the increased mutator phenotype in human HAP1 cells carrying the *hPMS1-S14P* or the *hMLH1-A21P* mutation. Future studies should evaluate whether a subset of the *hMLH1* and *hPMS2* missense mutations found in cancer patients are acting dominantly, similar as *MLH2* mutations identified in this work.

Clinically, in tumors displaying MSI without identified causative mechanism, i.e., absence of pathogenic mutations in *hMLH1*, *hMSH2*, *hMSH6*, or *hPMS2* and absence of *hMLH1* promoter methylation, the testing for somatic and germline mutations affecting *hPMS1* should be considered, particularly in patients with a clinical history suggestive of Lynch syndrome.

In summary, here we unraveled a novel mechanism that accounts for reduced MMR function in yeast and human cells, caused by dominant missense mutations in Mlh2 and hPMS1, respectively. Our results support a model in which specific Mlh2 mutations prevent the unloading of Mlh1–Mlh2 complexes from DNA, acting as roadblocks that interfere with downstream steps of the MMR reaction. Further studies are necessary to understand, at the molecular level, the biological process(es) affected by the mutations here identified. Such studies may shed light on specific residues that may participate in conformational changes driven by ATP binding/hydrolysis or interactions with DNA, MMR, or DNA replication components.

## Methods

**Yeast strains and media**. Strains used in this study (Supplementary Table 6) are derivatives of the S288c strain RDKY5964[20], with exception of strain AH109 (Clontech Laboratories) that was used for Y2H analysis. Strains were cultivated at 30 °C in yeast extract-peptone-dextrose media (YPD) or appropriate dextrose-containing synthetic dropout (SD) medium for selection of plasmids markers, lacking lysine (Lys−) or threonine (Thr−) (to select for *lys2-10A* or *hom3-10* frameshift revertants, respectively), or SD medium lacking arginine (Arg−) supplemented with 60 mg/L canavanine, to select canavanine-resistant (Can^R) mutants. 5-fluoroorotic acid (5-FOA, US Biological) plates were done in SD medium supplemented with 1 g/L 5-FOA. Antibiotics were used at the following final concentrations: 200 µg/mL geneticin (Santa Cruz Biotechnology), 300 µg/mL hygromycin B (Thermo Fisher Scientific), and 100 µg/mL nourseothricin (clonNAT, Werner BioAgents). Gene deletions and gene tagging were performed using standard PCR-based recombination methods[51,52], followed by confirmation by PCR. Tags and junctions were confirmed by PCR and sequencing. Yeast strains carrying mutations in *MLH1*, *MLH2*, or *PMS1* genes, were generated by pop-in/ pop-out strategy using pRS306-based integrative vectors[51], and were confirmed by sequencing.

**Construction of plasmids used in yeast experiments**. All plasmids used in this study are listed in Supplementary Table 7. The plasmid pHHB98 encodes the WT-*MLH2* gene, including 1 kb of the *MLH2* promoter and 300 bp of the terminator, cloned using the *SacI* and *EcoRI* sites in pRS316[51]. The *MLH2* sequence was amplified from genomic DNA using primers 5′-CTA CGA GAG CTC ACA AAT GGA TTC ATT AGA TCT ATT AC-3′ and 5′-GAG TAC GAA TTC TAT ATT

TAT GTG GAG TGA TCT TTG TC-3′. To generate pHHB157 (an integrative *URA3* plasmid containing the WT-*MLH2* gene), the *MLH2* gene (including promoter and terminator sequences) was cut from pHH98 with *SacI* and *EcoRI*, and ligated into pRS306. Integrative plasmids encoding specific mutant *mlh2* alleles were generated either by subcloning the mutant alleles identified in the screens or by site-directed mutagenesis. Integrative *mlh2* plasmids were linearized with *MluI* for integration at the *MLH2* locus.

The integrative plasmids pHHB270 (encoding the *mlh1-A18P* mutation) and pHHB240 (encoding the *pms1-S17P* specific mutation) were generated by site-directed mutagenesis of pRDK1338[25] and pRDK1667[25], respectively, followed by subcloning the mutant alleles at the *StuI* and *XhoI* sites in pRS306. pHHB270 and pHHB240 were linearized with *NheI* and *MluI*, respectively. The integrative plasmid pHHB252, encoding the *pol30-K217E* specific mutation, was generated by site-directed mutagenesis of the *LEU2*-integrative plasmid pRDK925[26]. pHHB252 was linearized with *SacI* for one-step replacement and Leu+ transformants were confirmed by sequencing.

**MLH2 random mutagenesis screen**. The plasmid library of randomly mutagenized *MLH2* was generated by mutagenic PCR amplification and in vivo gap repair in the *lig4Δ* strain HHY6620 that is deficient in non-homologous end joining, similar as previously described[22]. Transformants were plated on SD plates lacking uracil (Ura−) and then replica-plated onto frameshift mutator reporter plates (SD Ura− Lys− and SD Ura− Thr− to test for increased *lys2-10A* and *hom3-10* frameshift reversion mutations, respectively); and onto SD Ura− Arg− + 60 mg/L canavanine to identify *CAN1* inactivation mutations[8,32]. Plasmids resulting in increased frameshift mutations and canavanine resistance were identified, sequenced for *mlh2* mutations and retransformed in RDKY5964 for further analysis. To identify *mlh2-S16P*-dependent enhancer mutations, *mlh2-S16P* was randomly mutagenized by PCR and a screen was performed as described above.

**Determination of mutation rates in *S. cerevisiae***. Mutation rates using the *CAN1* inactivation assay and the frameshift reversion assays (*lys2-10A* and *hom3-10*) were determined by fluctuation analysis as previously described[8,32]. Each mutation rate was determined by using two biological isolates and at least 14 independent cultures. 95% confidence intervals were calculated for all fluctuation tests.

**Preparation of yeast and mammalian cell lysates and immunoblotting**. Whole-cell protein extracts of *S. cerevisiae* were generated as previously described[20]. Mammalian cell lysates were prepared in lysis buffer (10 mM Tris pH 7.5, 50 mM NaCl, 1% Triton X-100, 2 mM PMSF, and protease inhibitor cocktail Complete with EDTA (Roche). Lysates were analyzed on 8% or 10% SDS-PAGE followed by immunoblotting. The following antibodies were used: anti-HA (1:5,000, 3F10, Roche), anti-c-Myc (1:1,000, 4A6, Millipore), anti-Sic1 (1:10,000)[53], anti-Pgk1 (1:20,000, 22C5D8, Invitrogen), anti-hMLH1 (1:1,000, BD-551091), anti-hPMS1 (1:1,000, sc-615, Santa Cruz), and anti-actin (1:5,000, A2228, Sigma). Western blots were developed using Immobilon Western Chemiluminscent HRP substrate (Millipore) and imaged using Super RX-N Fuji medical X-ray films (Fujifilm) or using Fusion Solo S (Vilber).

**Live-cell imaging of Pms1 and Mlh2 foci in *S. cerevisiae***. To visualize the localization of Pms1 or Mlh2 proteins in living yeast cells we used strains expressing Pms1 or Mlh2 proteins tagged with a 4xGFP tag (a the C-terminus), strains which according to previous studies retain to large extent MMR proficiency[11,20]. Exponentially growing cells were processed and imaged as described[20] using a Leica SP5 confocal microscope system with an Argon laser, an HCX PL APO 63x/1.4 aperture objective and a high resonance scanner detector at 8 kHz frequency. 10–15 Z stacks spaced 0.4 µm were projected using the maximum intensity in ImageJ for analysis. Three independent biological replicates per genotype were analyzed and a Mann–Whitney rank sum test was used to compare Pms1/Mlh2 foci in different genetic backgrounds. Time-lapse images were performed in a DeltaVision RT (Applied Precision) with an inverted microscope (IX70, Olympus) with a CoolSNAP HQ2 (Photometrics) camera and a plan Apo 100x (1.4 NA) oil immersion objective lens (Olympus). 20 Z stacks spaced 0.3 µm were deconvolved using SoftWoRx software. For this experiment, exponentially growing cells (HHY8072 or HHY8073) grown in complete synthetic medium (CSM) were plated on glass chambers coated with concanavalin A (1 mg/ml, C-2010 Sigma) and imaged at 30 °C for indicated times. Deconvoluted images were processed to obtain the maximal intensity projection with ImageJ, at each shown time point. The Nup49-mCherry was used as nuclear marker. To compare Pms1 foci intensity between different genotypes, a culture of a strain expressing WT Pms1-4GFP (and Sik1-mCherry) (RDKY7600)[20] was mixed with culture of a mutant strain, e.g., *mlh2-S16P-D219G* Pms1-4GFP (HHY5326) in 3:1 ratio. Expression of the nucleolar marker Sik1 (tagged with mCherry) was used to identify WT Pms1-4GFP foci. GFP intensities were measured using ImageJ. Similarly, to measure Mlh2 foci intensity among strains with different genotypes, an *MLH2-4GFP NIC96-mCherry* strain (RDKY7905) was used mixed in a ratio 5:1 with a mutant strain (e.g., HHY8064 *mlh2-S16P-D219G-4GFP*).

**CAN1 mutation spectra analysis**. The CAN1 gene of at least 90 independent canavanine-resistant (CanR) yeast colonies was amplified from genomic DNA by PCR with Phusion high-fidelity DNA polymerase (Thermofisher) and primers 5′-GTT GGA TCC AGT TTT TAA TCT GTC GTC-3′ and 5′- TTC GGT GTA TGA CTT ATG AGG GTG-3′. PCR reactions were purified and sequenced (GATC Biotech). Sequences were analyzed with Lasergene 15.1 (DNASTAR).

**Yeast two-hybrid analysis**. Protein–protein interactions were tested using the Y2H system, using the strain AH109 (Clontech laboratories) (Supplementary Table 6). Briefly, AH109 strain was transformed with bait and prey plasmids, derivatives of pGBKT7 and pGADT7, respectively (Supplementary Table 7). Cells were grown overnight in SD media lacking tryptophan (Trp⁻) and leucine (Leu⁻) and were spotted in 10-fold serial dilutions on control plates (SD Trp⁻ Leu⁻) and reporter plates (SD Trp⁻ Leu⁻ His⁻ + 1 mM 3-amino-1,2,4-triazole (3-AT) (Sigma)). Plates were incubated at 30 °C for 3–4 days and imaged using GelDoc system (Bio-Rad). The expression of bait and prey proteins was confirmed by SDS-PAGE and immunoblotting using anti-HA and anti-Myc antibodies.

**Generation of plasmids used for CRISPR-Cas9 editing in mammalian cells**. Plasmids used in mammalian tissue culture experiments are listed in Supplementary Table 7. sgRNAs used for inactivation of the hMLH1 and hPMS1 genes or the introduction of hPMS1-S14P or hMLH1-A21P point mutations were designed with the Optimized CRISPR Design tool (http://crispr.mit.edu) and were cloned into pLentiCRISPR-puro (a gift from Feng Zhang, Addgene plasmid # 49535). This plasmid expresses a gene-specific sgRNA, the human codon-optimized Streptococcus pyogenes Cas9 protein and the puromycin N-acetyl transferase gene. The plasmid pHHB586 that contains the sgRNA-hMLH1 (5′-TGA TAG CAT TAG CTG GCC GC-3′) was used to target the hMLH1 gene. Plasmids pHHB484 that contains the sgRNA-hPMS1 (5′-TTC TCA GAT CAT CAC TTC GG-3′) and pHHB487 that contains the sgRNA-hPMS1 (5′-CAC AAG CGT AGA TGT TAA AC-3′) were used to target hPMS1. An sgRNA that targets GFP (5′-GGG CGA GGA GCT GTT CAC CG-3′) was similarly cloned into pLentiCRISPR-puro resulting in the plasmid pHHB761, which was used in control experiments (mock).

The donor plasmid pHHB730 that was used to introduce the hPMS1-S14P mutation contains a 5′ and 3′ homology arms of ~900 and ~300 bp long, respectively (relative to Ser14 coding sequence) (Supplementary Fig. 4). To construct pHHB730 a 1.3 kb DNA fragment of the hPMS1 gene (including part of 5′ UTR, exon 1, and part of a downstream intron) was PCR-amplified from genomic DNA isolated from HAP1 cells using primers (F) 5′ CTG ACT GGT ACC GTG CTT GTG GCA GAA TAT TGT GGA-3′ and (R) 5′ GGT GAC CTC GAG CCA TAT CTC TAT GTG TTA GCA-3′, and was cloned into pcDNA5 FRT/TO (Invitrogen) using KpnI and XhoI sites (underlined). This construct was then subjected to site-directed mutagenesis to introduce the S14P point mutation, a silent mutation (TGG to TCG) at the Protospacer adjacent motif (PAM) and a third silent mutation (TTACAA to TTATAA) that creates a PsiI cutting site ~100 bp upstream of the ATG start codon, used later for verification purposes. Finally, a hygromycin resistance cassette flanked by loxP sites was cloned at the intronic SwaI site located ~250 bp downstream of the Ser14 coding sequence in the donor plasmid. This hygromycin resistance cassette was obtained by PCR-amplification using pHHB689 as template and primers (F) 5′ CTGACT ATTTAAAT ATAACTTCGTATAGCATA CAT TAT ACG AAG TTA TAT ACG CGT TGA CAT TGA TTA TTG AC-3′ and (R) 5′- GGTGAC ATTTAAAT ATAACTTCGTATA ATG TAT GCT ATA CGA AGT TAT CAG AAG CCA TAG AGC CCA CC-3′ (SwaI sites underlined). Plasmid pHHB689 contains the hygromycin resistance gene cloned using the KpnI and XhoI sites in pcDNA5 FRT/TO (Invitrogen).

The hMLH1-A21P mutation was introduced into HAP1 cells using the donor plasmid pHHB690, which contains a 5′ and 3′ homology arms of ~1.2 kb and ~800 bp long, respectively (relative to Ala21 coding sequence) (Supplementary Fig. 5). The plasmid pHHB690 was constructed using a similar strategy as described for pHHB730. A ~2 kb fragment of the human hMLH1 gene (including part of the 5′ UTR, exon 1, and a downstream intron) was PCR-amplified from genomic DNA of HAP1 cells with primers (F) 5′-CTG ACT GGT ACC GCG TAG ATT CCT GTC AAT GCT CAG G-3′ and (R) 5′-GGT GAC CTC GAG CTC TCA GTC CCC TGA ATA G-3′ and was cloned using the BamHI and XhoI sites in pcDNA5 FRT/TO. Next, the A21P mutation and a silent mutation at the PAM (TGG to TGA, that creates an XmnI site used for verification purposes) were introduced by site-directed mutagenesis. Finally, a hygromycin resistance cassette flanked by loxP sites was introduced into an intronic SmaI site (136 bp downstream of the A21 coding sequence) in the donor plasmid. This loxP-flanked hygromycin gene resistance cassette was obtained by PCR amplification using the plasmid pHHB689 and the primers (F) 5′ CTG ACT CCCGGG ATAACTTCGTATA GCA TAC ATT ATA CGA AGT TAT ATA CGC GTT GAC ATT GAT TAT TGA C-3′ and (R) 5′ GGT GAC CCCGGG ATAACTTCGTATA ATG TAT GCT ATA CGA AGT TAT CAG AAG CCA TAG AGC CCA CC-3′, (SmaI sites are underlined).

**Generation of hMLH1-KO, hPMS1-KO, hPMS1-S14P, and hMLH1-A21P human cell lines**. hMLH1-KO cell lines were generated with the plasmid pHHB586. hPMS1-KO cell lines were generated with the plasmid pHHB484 and pHHB487.

An sgRNA-GFP encoded in the pHHB761 plasmid was used in control (Mock) experiments. Plasmids containing the sgRNAs were transfected into HAP1 cells by nucleofection using the Neon Transfection System (Life Technologies). Briefly, 10⁶ HAP1 cells were transfected with 2.5 μg of the corresponding sgRNA-carrying plasmid with 3 pulses at 1575 V for 10 ms. After transfection, cells were grown under puromycin selection (1 μg/mL) for 48 h, and then in the absence of the antibiotic for 10–12 days. Single clones were expanded and tested by Western blot for MLH1 expression and confirmed by sequencing analysis.

To generate HAP1 cells carrying the MLH1-A21P point mutation, cells were co-transfected (nucleofection) with pHHB586 and pHHB690 plasmids. HAP1 cells carrying the PMS1-S14P mutation were obtained after co-transfection with pHHB484 and pHHB730 plasmids. After 48 h post transfection, cells were selected with puromycin (1 μg/mL) for 48 h, recovered for 24 h, and finally selected with hygromycin (700 μg/mL, Sigma) for 10–12 days. The loxP-flanked hygromycin resistance cassette was excised by CRE-mediated site-specific recombination. Briefly, 10⁶ cells were transfected (nucleofection) with 2.5 μg CRE mRNA. Cells were grown without antibiotics for 10–12 days. Single clones were expanded and tested for loss of the hygromycin resistance. Finally, positive clones were identified by PCR and confirmed by sequencing analysis. All cells used in this study were negative for mycoplasma infection according to tests performed regularly (GATC, Eurofins Genomics).

**Hypoxanthine phosphoribosyl transferase 1 (HPRT1) inactivation assay**. HAP1 cells were usually grown in IMDM media supplemented with 10% FBS, in 5% CO₂ at 37 °C. However, for mutator phenotype analysis using the HPRT1 inactivation assay, cells were grown for 7–10 days (passaged every 48 h) in hypoxanthine-aminopterin-thymidine (HAT)-supplemented medium (Thermo Fisher Scientific) to purge pre-existing 6-TG resistant cells. After HAT treatment, cells were recovered for 2–3 passages. For the qualitative HPRT1 inactivation assay, 10⁴ cells of each clone were seeded in a 48-well plate in IMDM medium supplemented with 10% FBS. After 24 h, 6-TG was added and cells were grown for 7 days (medium was changed every 2–3 days). Finally, cells were fixed with ice-cold 20% methanol and were stained with 0.02% crystal violet (Sigma) in 20% methanol. A quantitative analysis of the 6-TG survival was determined with a colony formation assay by counting the number of 6-TG resistant colonies using four independent clones per genotype. Cells were plated at a density of 400 cells in 10 cm plates (in triplicate), in medium lacking 6-TG. After 24 h, 6-TG was added and cells were cultured for 12 days (medium was changed every 2–3 days). Plating efficiency was determined by plating 200 cells for each clone in 10 cm plates in media lacking 6-TG (in triplicate). At the end of the experiment, plates were stained, scanned and colonies were counted with ImageJ. The percentage of survival was calculated after counting the number of 6-TG resistant colonies (at each used concentration) and correcting for plating efficiency.

**Statistics and reproducibility**. To compare the percentage of nuclear Pms1/Mlh2 foci in different genetic backgrounds, three independent biological isolates per genotype were analyzed and a Mann–Whitney rank sum test was used for statistical analysis. P-values are indicated on the graphs and represent statistical significance of the difference between the two data groups. Mutation rates analysis in S. cerevisiae were determined using two independent biological isolates and a total of at least 14 independent cultures. The mutation rate data correspond to median rates for the indicated mutational reporters with 95% confidence intervals.

**Reporting summary**. Further information on research design is available in the Nature Research Reporting Summary linked to this article.

## Data availability
Source data for figures are provided in Supplementary data 1. Uncropped scans of Western blots are shown in Supplementary Fig. 7.

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

## Acknowledgements

We thank to Dr. Michael Knop for generous support and to Dr. Richard Kolodner for sharing strains, plasmids, reagents, and advice. We thank Dr. Christopher Putnam and Dr. Thomas Kunkel for helpful discussions. We thank Dr. Haikun Liu, Dr. Elmar Schiebel, Dr. Marius Lemberg, and Dr. Thomas Hofmann for sharing reagents and protocols, and the DKFZ Imaging Core Facility and Daniel Kirrmaier (Knop lab) for their technical support. We thank Umran Ceylan for initial Y2H interaction analysis. This work was supported by the Harald zur Hausen fellowship from the Deutsches Krebsforschungszentrum and the Deutsche Forschungsgemeinschaft DFG grant HO-5501/1-1 (both granted to H.H.).

## Author contributions

H.H. conceived in large the overall experimental design. G.X.R., T.T.S., K.G., B.Z., and H. H. performed yeast strain and plasmid construction, and quantitative mutation rate measurements. G.X.R. performed microscopy analysis of Pms1 and Mlh2 foci. B.Z. generated HAP1 mutant cells and performed mutator phenotype analysis. M.K. supervised MSI analysis in human cells. G.X.R., B.Z., and H.H. wrote the paper; and all of the authors revised and modified the paper.

## Funding

## Competing interests

The authors declare no competing interests.
