## [Peer Review File · Communications Biology]

Reviewers' comments:

Reviewer #1 (Remarks to the Author):

In this ms., the authors set out to uncover separation of function mutations in the yeast MLH2 gene and study the function of these mutations. By combining genetic studies, with cell and molecular biology studies, they gain insight into the mode of action of the Mlh1-Mlh2 complex in yeast showing that the mutated complex acts as a "roadblock" to affect DNA repair. They go on to create a mutant human version of the yeast mutation in cell lines and examine its effect on mutations.

Overall, the experiments described in this ms. are well-conceived and well-executed. The correct experiments are performed to address the questions that are posed and the authors clearly state their findings. I have only a couple negative criticisms. The authors need to explain their reasoning for stating in the first line of the discussion that the mutation is dominant? It was not clear to me that the yeast mutation was ever tested in a diploid. Furthermore, are they assuming that it is dominant since there is a wild type copy of MLH2 in the haploid? Please spell this problem out more clearly.

At the bottom of the 1st paragraph of the discussion, the authors reveal that the residue they uncovered in their yeast study, and then used to fashion the mutation in a mammalian cell, was an already known mutation from studies in E. coli. This previous observation somewhat diminishes the novelty of the current study.

There are a number of typos, etc.:

Abstract, ln. 7: phenotype needs to be singular – change to "Both aspects of this phenotype were..."

Results: Same for first title in the results section: "Identification of MLH2 dominant mutations resulting in a mutator phenotype."

To search...MLH2, for a dominant mutator phenotype. This plasmid library was transformed into a wild-type (WT) strain and transformants were screened for a mutator phenotype using... (added "a" 2 times)

Pg. 6, 8th ln. from bottom: Add comma: In some cases, a small increase in...

Pg. 12, ln. 4: ...associated to increased... -> ...associated with increased...

ln.5: ...specific mutations in hPMS1 gene... -> ...specific mutations in the hPMS1 gene

ln. 10: hPMS1-S14P mutation did not... -> The hPMS1-S14P mutation did not...

last ln.: 90% -> 85% (or whatever the percentage is, as it is not 90% in Figure 5c).

Pg. 14, ln. 5: ...is different to the one... -> ...is different from the one...

Pg. 16, ln. 4: ... similar as mutations preventing... -> ... similar to mutations preventing...

ln. 6: ...mutations are resulting in mutant... -> ...mutations result in mutant...

ln. 9-10: ...suppressed by a mutation that either prevent DNA binding (K294E) or inhibits ATP... -> ...suppressed by mutations that either prevent DNA binding (K294E) or inhibit ATP...

5th ln. from bottom: ...from DNA, process that... -> ...from DNA, a process that...

Last ln.: ...could be target of... -> ...could be targets of...

Pg. 18, 8th ln. from bottom: ...result in mutator phenotypes and cancer susceptibility. -> ...result in a mutator phenotype and cancer susceptibility.

Pg, 19, 1st ln.: Future studies might evaluate... -> Future studies should evaluate...

ln. 9: ...that gives account for... -> ...that accounts for...

4th ln. from bottom: Further studies will be necessary to understand at the molecular level the biological process(es) affected by the here identified mutations. These studies may... -> Further studies are necessary to understand, at the molecular level, the biological process(es) affected by the mutations identified here. Such studies may...

Pg. 25, 4th ln. from bottom: No space in restriction enzyme names and only italics for the first three letters - Sac I and EcoR I -> SacI and EcoRI - change throughout the ms..

Pg. 31, ln. 12: phosphorybosyl -> phosphoribosyl

Fig. 1. Change title: Identification of MLH2 dominant mutations causing mutator phenotypes. - > Identification of MLH2 dominant mutations causing a mutator phenotype.

Discussion, ln. 1: What is the reasoning for stating that the mutation is dominant? Was it ever tested in a diploid? Or at least with a wild type copy of MLH2 in the haploid?

Bottom of the 1st paragraph: The residue that they uncovered and then mutated in the mammalian cell was already known from studies in E. coli.

Reviewer #2 (Remarks to the Author):

Reyes et al. performed a screen to identify dominant *mlh2* mutations that disrupt DNA mismatch repair (MMR). They identified several alleles that map to the N-terminal ATP binding domain of Mlh2. The mutations confer a strong mutator phenotype when expressed alone (~1000-fold compared to wild-type, in an assay in which the *msh2null* displays an ~7000-fold elevated mutation rate), or when expressed in the presence of wild-type MLH2 (though a bit weaker). Their work is consistent with the *mlh2* mutations locking the Mlh1-Mlh2 complex in a state that prevents their N-terminal ATP binding domains from dissociating. The authors hypothesize that such a locked complex prevents the major MLH mismatch repair complex, Mlh1-Pms1, from being able to act in mismatch repair. This idea is supported by the finding that MLH foci are longer lasting in *mlh2* dominant mutation backgrounds and that the dominant *mlh2* mutations disrupt both Exo1-dependent and independent MMR.

Comments

This work follows from previous studies which identified dominant mutations in MMR genes that disrupted multiple MMR pathways. For example, Das Gupta and Kolodner (Nat Genet 24:53) identified *msh6* mutations that appear to inactivate both Msh2-Msh6 and Msh2-Msh3 pathways, and subsequent biochemical analysis (Hess et al. PNAS 103:558) indicated that the mutant Msh2-msh6 complex could assemble ternary complexes with Mlh1-Pms1 but could not form ATP-induced sliding clamps. While it does not appear that such *mlh2* mutations are common in cancer databases, the finding that a complex that plays a very minor role in mismatch repair (in repairing frameshift mutations at a low efficiency), but when mutated can have a severe defect in a major pathway, is interesting.

This work is very well executed and is also of interest in terms of obtaining a possible explanation/model for phenotypes in tumors that display microsatellite instability but do not have mutations in key mismatch repair proteins. It would have been nice to see some of the mechanistic work done previously (e.g. does Mlh1-mlh2 form ternary complexes that are defective?), but I don't think of such experiments as being critical. I have some concerns about controls that are listed below.

1. Line 103, 278. Figure 5. The human cell line experiments do not appear to have included a hPms1^{-/-} control. While such a cell line is likely to be a weak mutator based on other studies, it is still important to include such a control.

2. Line 248, Table 2. Is there a reason that the authors did not test overexpression of Mlh2? Such a control would test if such overexpression alone confers a dominant negative phenotype. This could be another explanation (albeit unlikely) for why the mlh2 alleles show a dominant negative phenotype (due to increased protein stability) and why the phenotype appears strongest in an assay which primarily measures frameshift mutations.

3. Line 256. It's interesting to see that overexpression of POL30 suppressed the mutator phenotype seen in the mlh2 dominant mutant strains. This might suggest that Pol30 physically interacts with Mlh2. Is there any evidence for this?

4. Line 678. I may have missed this, or perhaps the information can be found in an earlier paper. Did the authors test if the Pms1 and Mlh2 4xGFP tagged proteins complement their respective null mutations? This information should be included somewhere.

5. The paper feels long relative to the observations that were made. Some ideas to shorten the paper are:

-Tables 1 and 2 are redundant with Figure 2C, 4B, C, D. One could show the mutation rate data in the tables and eliminate the above figure panels.

-Tables 1 and 2 can be combined into a single table. As an aside, I'm curious why a pms1 null is not shown in Table 1.

Minor Comments

Line 28 and elsewhere. It would be worth indicating to the reader the significance of a 1000-fold increase in mutation rate. The assay in question displays a 7000-fold range, and when using less-sensitive assays the phenotype is appears less dramatic. One solution is to edit the text to say: "...rates up to 1000-fold in an assay in which a MMR defective strain shows 7000-fold higher rates compared to wild-type."

Line 85. "We have previously shown..." should be changed to "Campbell et al. (2014) previously showed..."

Reviewer #3 (Remarks to the Author):

Inactive mutations of the main MMR factors lead to hyper-mutations across species and underlie the lynch syndrome in humans. Here, the authors uncover dominant mutations of an MMR factor MLH2 in yeast and its ortholog hPMS1 in humans, which normally play minor roles during mutation correction. Using a plasmid-based screen in yeast, four Mlh2 mutations located its NTD were found to be dominant mutators. Integrated Mlh2 alleles did not change protein levels but had dominant effects on both mutation rates and persistence of Psm1 foci levels. Focusing on mlh2-S16P, the authors uncover six mutations in MLH2 NTD that enhance its mutator effects (ave ~12X), increase Mlh2 foci levels and brightness, and Pms1 loci duration (ave ~8X). The mlh2-S16P-D219D double mutant increases Mlh2 protein levels and also increase Mlh1 interaction in yeast two-hybrid assays. The latter effect was shown to be due to S16P alone though. Epistatic analyses suggest that mlh2-S16P-D219D or mlh2-S16P effect bot exo1 dependent and independent steps acting downstream of MLH. Mutation blocking Mlh2 binding to DNA (K294E) or ATP hydrolysis (E29A) eliminate the mlh2-S16P-D219D mutator phenotype and increased levels of Mlh2 foci without affect protein levels. Similar suppression was seen with PCNA and PMS1 overexpression.

Interestingly, homolog mutations in PMS1 and Mlh1 lead to mutator phenotype, suggesting the conserved residues among these three proteins is important. The authors found that equivalent mutation in hPMS1 lead to increase mutations. This reviewer found the study is well done with high quality of data and interesting new MLH2/hPMS1 mutations that act dominantly. These mutations can provide clues for how different MLH complexes function during their ATP cycle. I only have two small questions that could further enhance this already nice work.

The rescue of mlh2-S16P-D219D defect by K294E and E29A mutations is striking. Do the triple mutants still able to bind to Mlh1 in yeast two hybrid?

The rescue of mlh2-S16P-D219D defect by PCNA OE is interesting. While the authors provide a reasonable genetic interpretation, the biochemical mechanisms of the suppression can be more clarified. For example, could PCNA OE reserve the enhanced Mlh1 interaction with mlh2-S16P-D219D mutant proteins?

Reviewer #1 (Remarks to the Author):

In this ms., the authors set out to uncover separation of function mutations in the yeast MLH2 gene and study the function of these mutations. By combining genetic studies, with cell and molecular biology studies, they gain insight into the mode of action of the Mlh1-Mlh2 complex in yeast showing that the mutated complex acts as a “roadblock” to affect DNA repair. They go on to create a mutant human version of the yeast mutation in cell lines and examine its effect on mutations.

Overall, the experiments described in this ms. are well-conceived and well-executed. The correct experiments are performed to address the questions that are posed and the authors clearly state their findings. I have only a couple negative criticisms. The authors need to explain their reasoning for stating in the first line of the discussion that the mutation is dominant? It was not clear to me that the yeast mutation was ever tested in a diploid. Furthermore, are they assuming that it is dominant since there is a wild type copy of MLH2 in the haploid? Please spell this problem out more clearly.

Thanks for this opportunity to clarify this point. The identification of *mlh2* mutations resulting in elevated mutation rates was done in haploid strains expressing the *mlh2* mutant alleles in a low copy number plasmid, but carrying a WT-*MLH2* chromosomal allele (Supplementary Table 1). In addition, we reported mutation rates in haploid strains in which we replaced the wild-type *MLH2* gene with the *mlh2* mutant alleles at the chromosomal locus (Table 1 and 2). The fact that deleting the *MLH2* gene in *S. cerevisiae* causes a very mild mutator phenotype (if any) (PMID:10679328, PMID: 24811092, and Table 1), indicates that this protein is largely dispensable for MMR. Therefore, the mutator phenotype associated to the *mlh2* mutations identified here is not caused by the loss of Mlh2 function, but rather as consequence of Mlh2 mutant complexes that interfere in a dominant manner with the function of other more relevant MMR proteins. Our experiments support a model in which Mlh1-Mlh2 mutant complexes are acting as roadblocks on DNA, preventing the downstream steps of the MMR reaction (Mlh1-Pms1 endonuclease reaction and Exo1-dependent excision).

We have modified the first paragraph of the results and the discussion sections to state more clearly how these *mlh2* mutations confer a dominant mutator phenotype (line 113; and lines 335-339).

At the bottom of the 1st paragraph of the discussion, the authors reveal that the residue they uncovered in their yeast study, and then used to fashion the mutation in a mammalian cell, was an already known mutation from studies in *E. coli*. This previous observation somewhat diminishes the novelty of the current study.

I would rather say that this similarity reinforces the idea that the here-identified mutations might affect important functions that have been conserved across MutL homologs in different species. Moreover, the previous study that identified MutL dominant negative mutations in *E. coli* (PMID: 8692687) reported a variety of additional mutations that were not found in this study, despite the fact that our screen was largely saturated as several of the *mlh2* mutations were identified more than once. Moreover, we tested some of these EcMutL dominant mutations (e.g. E29K, D58N, R95C and G96D) in yeast by mutating the homolog residue in Mlh2, and found that none of those tested caused a mutator phenotype (data not shown). These differences could be related to the fact that in *E. coli* there is only one homodimeric MutL complex, whereas in eukaryotes there are up to three heterodimeric MutL complexes. In the revised version of our manuscript, we are taking in account this issue at the end of the first paragraph in the discussion (page 15, lines 352-357).

There are a number of typos, etc.:

Abstract, ln. 7: phenotype needs to be singular – change to “Both aspects of this phenotype were...”

corrected.

Results: Same for first title in the results section: “Identification of MLH2 dominant mutations resulting in a mutator phenotype.”

To search...MLH2, for a dominant mutator phenotype. This plasmid library was transformed into a wild-type (WT) strain and transformants were screened for a mutator phenotype using... (added “a” 2 times)

corrected.

Pg. 6, 8th ln. from bottom: Add comma: In some cases, a small increase in...

done.

Pg. 12, ln. 4: ...associated to increased... -> ...associated with increased...

done

ln.5: ...specific mutations in hPMS1 gene... -> ...specific mutations in the hPMS1 gene

corrected.

ln. 10: hPMS1-S14P mutation did not... -> The hPMS1-S14P mutation did not...

done.

last ln.: 90% -> 85% (or whatever the percentage is, as it is not 90% in Figure 5c).

We apologize for this oversight. The percentage corresponds to 82% (corrected in page 14, line 320).

Pg. 14, ln. 5: ...is different to the one... -> ...is different from the one...

corrected.

Pg. 16, ln. 4: ... similar as mutations preventing... -> ... similar to mutations preventing...

done.

ln. 6: ...mutations are resulting in mutant... -> ...mutations result in mutant...

corrected.

ln. 9-10: ...suppressed by a mutation that either prevent DNA binding (K294E) or inhibits ATP... -> ...suppressed by mutations that either prevent DNA binding (K294E) or inhibit ATP...

done.

5th ln. from bottom: ...from DNA, process that... -> ...from DNA, a process that...

corrected.

Last ln.: ...could be target of... -> ...could be targets of...

corrected.

Pg. 18, 8th ln. from bottom: ...result in mutator phenotypes and cancer susceptibility. -> ...result in a mutator phenotype and cancer susceptibility.

done.

Pg, 19, 1st ln.: Future studies might evaluate... -> Future studies should evaluate...

corrected.

ln. 9: ...that gives account for... -> ...that accounts for...

done.

4th ln. from bottom: Further studies will be necessary to understand at the molecular level the biological process(es) affected by the here identified mutations. These studies may... -> Further studies are necessary to understand, at the molecular level, the biological process(es) affected by the mutations identified here. Such studies may...

changed as indicated.

Pg. 25, 4th ln. from bottom: No space in restriction enzyme names and only italics for the first three letters - Sac I and EcoR I -> SacI and EcoRI – change throughout the ms..

done.

Pg. 31, ln. 12: phosphorybosyl -> phosphoribosyl

corrected.

Fig. 1. Change title: Identification of MLH2 dominant mutations causing mutator phenotypes. -> Identification of MLH2 dominant mutations causing a mutator phenotype.

done.

Discussion, ln. 1: What is the reasoning for stating that the mutation is dominant? Was it ever tested in a diploid? Or at least with a wild type copy of MLH2 in the haploid?

The *mlh2* mutations identified here result in a mutator phenotype in the presence of a wild-type copy of *MLH2* gene in a haploid strain (results presented in Supplementary Table 1); additional tests in a diploid background were not performed. As mentioned at the beginning (please read our reply to the first comment raised by this reviewer), these *mlh2* mutations cause a dominant mutator phenotype because they interfere with the function of other more relevant components of the MMR pathway.

Bottom of the 1st paragraph: The residue that they uncovered and then mutated in the

mammalian cell was already known from studies in E. coli.

This point was already addressed, please see comment above.

Reviewer #2 (Remarks to the Author):

Reyes et al. performed a screen to identify dominant mlh2 mutations that disrupt DNA mismatch repair (MMR). They identified several alleles that map to the N-terminal ATP binding domain of Mlh2. The mutations confer a strong mutator phenotype when expressed alone (~1000-fold compared to wild-type, in an assay in which the msh2null displays an ~7000-fold elevated mutation rate), or when expressed in the presence of wild-type MLH2 (though a bit weaker). Their work is consistent with the mlh2 mutations locking the Mlh1-Mlh2 complex in a state that prevents their N-terminal ATP binding domains from dissociating. The authors hypothesize that such a locked complex prevents the major MLH mismatch repair complex, Mlh1-Pms1, from being able to act in mismatch repair. This idea is supported by the finding that MLH foci are longer lasting in mlh2 dominant mutation backgrounds and that the dominant mlh2 mutations disrupt both Exo1-dependent and independent MMR.

Comments

This work follows from previous studies which identified dominant mutations in MMR genes that disrupted multiple MMR pathways. For example, Das Gupta and Kolodner (Nat Genet 24:53) identified msh6 mutations that appear to inactivate both Msh2-Msh6 and Msh2-Msh3 pathways, and subsequent biochemical analysis (Hess et al. PNAS 103:558) indicated that the mutant Msh2-msh6 complex could assemble ternary complexes with Mlh1-Pms1 but could not form ATP-induced sliding clamps. While it does not appear that such mlh2 mutations are common in cancer databases, the finding that a complex that plays a very minor role in mismatch repair (in repairing frameshift mutations at a low efficiency), but when mutated can have a severe defect in a major pathway, is interesting.

This work is very well executed and is also of interest in terms of obtaining a possible explanation/model for phenotypes in tumors that display microsatellite instability but do not have mutations in key mismatch repair proteins. It would have been nice to see some of the mechanistic work done previously (e.g. does Mlh1-mlh2 form ternary complexes that are defective?), but I don't think of such experiments as being critical. I have some concerns about controls that are listed below.

1. Line 103, 278. Figure 5. The human cell line experiments do not appear to have included a hPms1-/- control. While such a cell line is likely to be a weak mutator based on other studies, it is still important to include such a control.

We apologize that did not include this control in our first submission, despite the fact that we had already compared side-by-side hPMS1-KO, hPMS1-S14P and the parental HAP1 cells (mock) cells with the HPRT inactivation assay. In this revised version we included Suppl. Fig. 6, that shows that the inactivation of the *hPMS1* gene does not confer an increased survival to 6-TG (compared to HAP1-mock cells), whereas HAP1 cells expressing the *hPMS1-S14P* mutation show at least a 2-fold higher survival at all tested 6-TG concentrations, compared to HAP1-mock cells. These results are commented on page 13 (lines 301-304; page 14, lines 320-323).

2. Line 248, Table 2. Is there a reason that the authors did not test overexpression of Mlh2? Such a control would test if such overexpression alone confers a dominant negative phenotype. This could be another explanation (albeit unlikely) for why the mlh2 alleles show a dominant negative phenotype (due to increased protein stability) and why the phenotype appears strongest in an assay which primarily measures frameshift mutations.

As mentioned in the introduction (page 4, last paragraph), Campbell et al 2014 has previously shown that *MLH2* overexpression (and also *MLH3-OE*) completely abolished MMR function (PMID: 24811092), and therefore it was important to find out if the *mlh2* mutations identified in this screen affected or not Mlh2 protein levels. For this reason we introduced the identified *mlh2* mutations into the *MLH2* chromosomal locus and compared Mlh2 protein expression levels with a wild-type strain. Results shown in Suppl. Fig 1 and Suppl. Fig 2 showed that the *mlh2* mutations are not causing major changes in Mlh2 protein levels. Therefore, we hypothesized that these mutant protein complexes are compromising the MMR function by a different mechanism. Additional experiments presented along the manuscript sustain a model in which Mlh1-Mlh2 mutant complexes act as roadblocks on DNA preventing the function of other more relevant MMR proteins (e.g. Mlh1-Pms1 and Exo1).

3. Line 256. It's interesting to see that overexpression of POL30 suppressed the mutator phenotype seen in the *mlh2* dominant mutant strains. This might suggest that Pol30 physically interacts with Mlh2. Is there any evidence for this?

We do not have experimental evidence that supports a physical interaction between Pol30 and Mlh2 (mainly, based on our own negative results obtained from yeast two hybrid experiments). However, it is known that Pol30 interacts transiently with the C-terminus of Mlh1-Pms1 (PMID: 28439008) in order to activate the Mlh1-Pms1 endonuclease. Since some of the residues involved in this interaction are part of Mlh1, it is possible that PCNA will also interact with Mlh1-Mlh2 complexes. However, the accumulation of Pms1/Mlh2 foci in strains expressing *mlh2* dominant mutations strongly argues that these mutations are not preventing the recruitment of these complexes at the mispair site, but rather are somehow interfering with their turnover.

4. Line 678. I may have missed this, or perhaps the information can be found in an earlier paper. Did the authors test if the Pms1 and Mlh2 4xGFP tagged proteins complement their respective null mutations? This information should be included somewhere.

We apologize for this oversight. Yes, in a previous report it has been shown that the tagged proteins retain their functionality. We have modified the method section where we refer to "Live-cell imaging of Pms1 and Mlh2 foci in *S. cerevisiae*" accordingly (page 23, lines 564-568).

5. The paper feels long relative to the observations that were made. Some ideas to shorten the paper are:

-Tables 1 and 2 are redundant with Figure 2C, 4B, C, D. One could show the mutation rate data in the tables and eliminate the above figure panels.

-Tables 1 and 2 can be combined into a single table. As an aside, I'm curious why a *pms1* null is not shown in Table 1.

I totally agree that panels 1D, 2C, 4B-D are redundant with information presented in Table 1 and 2. However, I believe that these graphs (that now have been converted in box-plots to adhere to journal formatting instructions) are for most readers (especially those from other research fields) more amenable than the tables. On the other hand, the tables contain more detailed information (including additional mutator assays) that will be appreciated by scientists more closely related to our field. In this revised version we have converted Table 1 and 2 into Supplementary Tables 1 and 2. I hope this modification is acceptable.

In regard the second point, I do not understand the need to show a *pms1* null strain. Previous reports have already shown that a *pms1* null strain is equivalent to an *msh2* null or an *mlh1* null strain (PMID: 8371783, PMID: 8666228). Perhaps it was meant *mlh2* null, but the reviewer was confused with the differences in MMR gene-name nomenclature between human and yeast? (Yeast Pms1 is the homolog of human PMS2, whereas yeast Mlh2 is the homolog of human Pms1).

In Table 1 (now called Supplementary Table 2) we have used an *msh2* null strain as a MMR deficient control strain. In addition, this table includes an *mlh2* null strain that causes a very modest mutator phenotype compared to the here identified *mlh2* dominant mutations.

Minor Comments

Line 28 and elsewhere. It would be worth indicating to the reader the significance of a 1000-fold increase in mutation rate. The assay in question displays a 7000-fold range, and when using less-sensitive assays the phenotype is appears less dramatic. One solution is to edit the text to say: "...rates up to 1000-fold in an assay in which a MMR defective strain shows 7000-fold higher rates compared to wild-type."

Due to space limitations in the abstract we have deleted the statement of "up to 1000-fold", but we have modified the text later on as suggested (page 5, lines 99-101).

Line 85. "We have previously shown..." should be changed to "Campbell et al. (2014) previously showed...."

This change has been made as suggested (page 4, line 85 and 88).

Reviewer #3 (Remarks to the Author):

Inactive mutations of the main MMR factors lead to hyper-mutations across species and underlie the lynch syndrome in humans. Here, the authors uncover dominant mutations of an MMR factor MLH2 in yeast and its ortholog hPMS1 in humans, which normally play minor roles during mutation correction. Using a plasmid-based screen in yeast, four Mlh2 mutations located its NTD were found to be dominant mutators. Integrated Mlh2 alleles did not change protein levels but had dominant effects on both mutation rates and persistence of Pms1 foci levels. Focusing on *mlh2*-S16P, the authors uncover six mutations in MLH2 NTD that enhance its mutator effects (ave ~12X), increase Mlh2 foci levels and brightness, and Pms1 loci duration (ave ~8X). The *mlh2*-S16P-D219D double mutant increases Mlh2 protein levels and also increase Mlh1 interaction in yeast two-hybrid assays. The latter effect was shown to be due to S16P alone though. Epistatic analyses suggest that *mlh2*-S16P-D219D or *mlh2*-S16P effect both *exo1* dependent and independent steps acting downstream of MLH. Mutation blocking Mlh2 binding to DNA (K294E) or ATP hydrolysis (E29A) eliminate the *mlh2*-S16P-D219D mutator phenotype and increased levels of Mlh2 foci without affect protein levels. Similar suppression was seen with PCNA and PMS1 overexpression. Interestingly, homolog mutations in PMS1 and Mlh1 lead to mutator phenotype, suggesting the conserved residues among these three proteins is important. The authors found that equivalent mutation in hPMS1 lead to increase mutations. This reviewer found the study is well done with high quality of data and interesting new MLH2/hPMS1 mutations that act dominantly. These mutations can provide clues for how different MLH complexes function during their ATP cycle. I only have two small questions that could further enhance this already nice work.

The rescue of *mlh2*-S16P-D219D defect by K294E and E29A mutations is striking. Do the triple mutants still able to bind to Mlh1 in yeast two hybrid?

Thanks for this interesting suggestion. We have generated the suggested triple mutant plasmids and have repeated the yeast two hybrid (Y2H), now including these two triple mutants (shown in the new Fig. 4A), and the western blot analysis (new Supplementary Fig. 3A). These additional experiments revealed that the E29A mutation does not cause significant changes in the Y2H interaction between the *mlh2*-S16P-D219G and Mlh1 proteins. In contrast, the K294E mutation that is predicted to prevent DNA binding, completely abolished their interaction by Y2H. It is interesting to point out that out of the E29A and the K294E mutations, the latter resulted in a more

efficient suppression of the dominant mutator phenotype (Table 2B, now renamed as Supplementary Table 3) and the accumulation of Mlh2 foci (Supplementary Fig. 3B) associated to the *mlh2-S16P-D219G* mutation. These results are discussed in page 11, lines 251-260.

The rescue of mlh2-S16P-D219D defect by PCNA OE is interesting. While the authors provide a reasonable genetic interpretation, the biochemical mechanisms of the suppression can be more clarified. For example, could PCNA OE reserve the enhanced Mlh1 interaction with mlh2-S16P-D219D mutant proteins?

Yes, it is an interesting possibility that should be tested in the future. The current MMR model proposes that after recruitment of Mlh1-Pms1 at the mismatch site, PCNA transiently interacts and activates Mlh1-Pms1 endonuclease activity. Despite the lack of experimental evidence at this point, one possibility is that this interaction triggers a conformational change in Mlh1-Pms1 (and potentially also Mlh1-Mlh2) that facilitates the unloading of these complexes from DNA. This hypothesis is consistent with the accumulation of Pms1 foci reported in yeast strains carrying PCNA mutant alleles that are defective activating the Mlh1-Pms1 endonuclease (PMID: 24981171), but also with the suppression of *mlh2-S16P-D219G* mutator phenotype and the reduction in Mlh2 foci abundance observed upon Pol30-OE. We have added a paragraph in the discussion where we comment on this possibility (page 12, lines 272-278).

REVIEWERS' COMMENTS:

Reviewer #1 (Remarks to the Author):

The authors have adequately addressed my concerns.

Reviewer #2 (Remarks to the Author):

The authors have fully addressed all of my comments.

Reviewer #3 (Remarks to the Author):

The authors have addressed all my comments. Congratulations for a nice work!

Manuscript COMMSBIO-20-1678A

Point-by-point response to reviewer's comments

Reviewer #1 (Remarks to the Author):

The authors have adequately addressed my concerns.

Reviewer #2 (Remarks to the Author): The authors have fully addressed all of my comments.

Reviewer #3 (Remarks to the Author): The authors have addressed all my comments. Congratulations for a nice work!

We thank all three reviewers for the positive assessment and the recommendation to publish our work in Communications Biology.